# Mechanistic neutral models show that sampling biases drive the apparent explosion of early tetrapod diversity

Emma M. Dunne [1,2,6] ✉, Samuel E. D. Thompson [3,4,6], Richard J. Butler [2,7],
James Rosindell [3,7] & Roger A. Close [5,7]

Estimates of deep-time biodiversity typically rely on statistical methods to mitigate the impacts of sampling biases in the fossil record. However, these methods are limited by the spatial and temporal scale of the underlying data. Here we use a spatially explicit mechanistic model, based on neutral theory, to test hypotheses of early tetrapod diversity change during the late Carboniferous and early Permian, critical intervals for the diversification of vertebrate life on land. Our simulations suggest that apparent increases in early tetrapod diversity were not driven by local endemism following the 'Carboniferous rainforest collapse'. Instead, changes in face-value diversity can be explained by variation in sampling intensity through time. Our results further demonstrate the importance of accounting for sampling biases in analyses of the fossil record and highlight the vast potential of mechanistic models, including neutral models, for testing hypotheses in palaeobiology.

The establishment of terrestrial ecosystems and diversification of early tetrapods during the late Carboniferous and early Permian (323–272 million years ago (Ma)) was a key event in vertebrate evolution. This interval was punctuated by a climate change-driven floral turnover at the end of the Carboniferous, referred to as the 'Carboniferous rainforest collapse' (CRC)[1,2]. In the past decade, several studies have attempted to estimate the impact of the CRC on the diversity of early tetrapods (early representatives of amphibians and amniotes). The first investigation into the impact of the CRC on early tetrapod diversification, by Sahney et al.[3], hypothesized that habitat fragmentation caused by the CRC drove increased endemism in early tetrapod communities via the 'island-biogeography effect', causing allopatric speciation in newly isolated patches of forest[4]. Such increases in local endemism would, in turn, be expected to lead to a rise in beta diversity and global species richness, coupled with a decline in local richness (alpha diversity)[3]. This interpretation has been challenged, however, because it took the fossil record at face value and thus did not compensate for pervasive biases

caused by various interconnected geological, taphonomic, anthropogenic and historical factors, which result in an uneven spatial and temporal distribution of fossil occurrences[3–7].

More recent investigations have pointed to sampling biases as a possible alternative explanation, and find no evidence of increases in endemism during or after the CRC[5,6]. In particular, Dunne et al.[6], after correcting for sampling, found evidence of increased connectedness between early tetrapod communities (for both amphibians and amniotes) and lower 'global' diversity following the CRC—the opposite of that reported by Sahney et al.[3]. The same study[6] also suggested that fragmentation of the rainforest probably promoted the recovery and subsequent diversification of amniotes, a clade that today comprises reptiles, birds and mammals[6]. Despite these advances, the early tetrapod fossil record remains fragmentary as well as unevenly and incompletely sampled (particularly in the late Carboniferous)[7], which obscures patterns of diversity and biogeography during a critical time in vertebrate evolution[6]. The true joint effects of sampling and

¹GeoZentrum Nordbayern, Friedrich-Alexander University Erlangen-Nürnberg (FAU), Erlangen, Germany. ²School of Geography, Earth and Environmental Sciences, University of Birmingham, Birmingham, UK. ³Department of Life Sciences, Imperial College London, Ascot, UK. ⁴Department of Biological Sciences, National University of Singapore, Singapore, Singapore. ⁵Department of Earth Sciences, University of Oxford, Oxford, UK. ⁶These authors contributed equally: Emma M. Dunne, Samuel E. D. Thompson. ⁷These authors jointly supervised this work: Richard J. Butler, James Rosindell, Roger A. Close. ✉e-mail: dunne.emma.m@gmail.com

## a

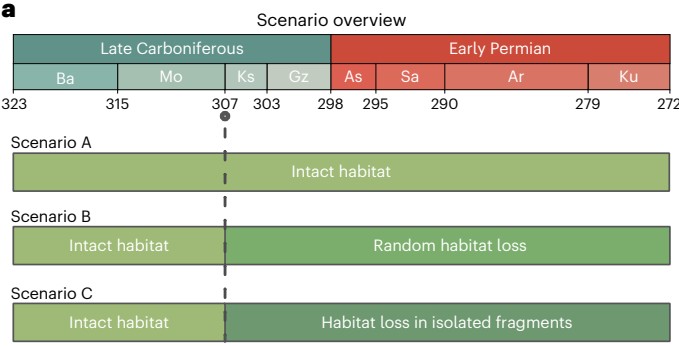

## b

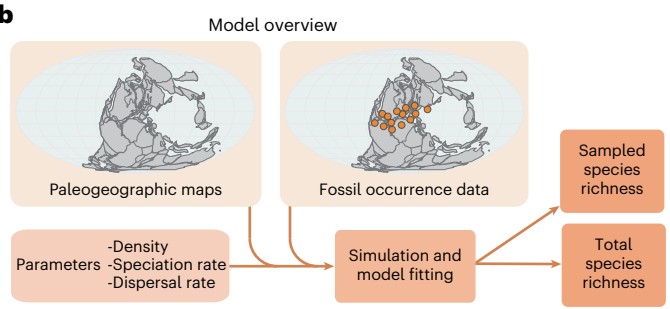

**Fig. 1 | Schematic outlining the study methodology. a,** Visualization of the three simulated scenarios related to the CRC. Scenario A performs simulations on a 'pristine' landscape with no habitat fragmentation (that is, the CRC was absent from this scenario). Scenario B models the effect of the CRC as random habitat loss across the landscape. Scenario C models the effect of the CRC as a loss in habitat in which 'habitat islands' remained around the localities where early tetrapods occurred. **b,** An overview of the model input data, parameters and predicted outputs. Ar, Artinskian; As, Asselian; Ba, Bashkirian; Gz, Gzhelian; Ks, Kasimovian; Ku, Kungurian; Mo, Moscovian; Sa, Sakmarian.

environmental change on the diversification of tetrapods following the CRC are yet to be unravelled.

Quantitative approaches to correcting for the effects of sampling biases on estimates of past biodiversity typically rely on statistical or phylogenetic techniques[8,9]. These approaches have led to substantial revisions of diversity patterns in many fossil groups, including early tetrapods[10–15].

Mechanistic neutral models provide an alternative and complementary approach that has not yet been used widely in palaeobiological studies (but see ref. 16). Neutral models assume individual dynamics are independent of species identity. Making this strong assumption puts the focus on sampling, habitat structure and dispersal in isolation from other potential complicating factors. It also permits use of particularly efficient simulation algorithms[17], enabling us to study spatially explicit samples of individuals from a very large spatially explicit landscape that would be impractical to simulate mechanistically under alternative models. Crucially, such neutral simulations can specify landscapes with realistic size and structure, and enable features such as palaeogeography, habitat loss and habitat fragmentation to be manipulated experimentally[18]. Diversity can then be sampled from the simulations in the same locations and with the same intensity as the empirical data, thus providing a new way to test how real-world patterns of fossil record sampling impact inferred patterns of face-value (directly observed, 'raw' or uncorrected) diversity, under a range of hypothetical palaeogeographic or ecological scenarios. The mechanistic nature of neutral models also enables us to run the models with samples much larger than the empirical sample sizes. This allows us to predict how observed diversity patterns might change if the intensity of fossil sampling were increased by an order of magnitude, and to understand what patterns could be detected within the currently

available fossil data[19]. Studies involving neutral simulations can even be used to test theories of diversity dynamics at global scales, much larger than could be directly quantified in the fossil record because of incomplete spatial sampling[16,20,21].

In this study, we apply a spatially explicit version of neutral theory to test the hypothesis that the CRC impacted early tetrapod diversity through habitat fragmentation. Our neutral simulations mimic the empirical structure of the fossil record by sampling at the same locations and to approximately the same intensity as recorded in the empirical or 'known' early tetrapod data. We investigate three scenarios related to the CRC. The first (scenario A; Fig. 1a) performs simulations on a 'pristine' landscape with no habitat fragmentation (that is, the CRC was absent from this scenario). The second (scenario B; Fig. 1a) models the effect of the CRC as random habitat loss across the landscape. The final scenario (C; Fig. 1a) models the effect of the CRC as a loss in habitat in which 'habitat islands' remained around the localities where early tetrapods occurred. We use these spatially explicit neutral simulations to examine the extent to which the empirical fossil record, given its sampling bias, can infer global patterns of diversity change. We estimate trends in tetrapod diversity over time, under a neutral scenario, by simulating our best-fitting models again with constant temporal sampling. This study is, to the best of our knowledge, the first to apply a fully spatially explicit neutral model to empirical fossil data.

## Results

### Neutral models incorporating temporal and spatial biases

In the simplest scenario, we performed simulations on a pristine global landscape with no habitat fragmentation (scenario A; Fig. 1). We sampled diversity patterns from the simulations at the same palaeo-locations and to approximately the same intensity as the real fossil record. Results from the simulations with optimal fixed parameters matched the empirical fossil record well overall, with 80–85% mean accuracy across four diversity metrics: alpha diversity across all localities, mean alpha diversity, beta diversity and gamma diversity (Methods). The simulations could not, however, reproduce alpha, beta and gamma diversity well with the same fixed set of parameters. When compared with the empirical fossil record, the neutral models of early Permian communities with optimized fixed parameters predicted more species (higher gamma diversity), and higher alpha diversity than seen empirically (Fig. 2). Despite these differences, the majority of the empirical values are within the range of variation between simulations, with a mean accuracy across all metrics of 81.3% for amniotes and 82.1% for amphibians.

### Neutral models under habitat fragmentation

To test the hypothesis that fragmentation of the rainforest at the end of the Carboniferous promoted the development of endemism among early tetrapod communities, we modelled two scenarios of habitat loss and fragmentation occurring from 307 Ma onwards: first, a random pattern of habitat loss (scenario B; Fig. 1) and second, a clustered pattern of habitat loss (scenario C; Fig. 1). The random habitat loss scenario (B) maintains connectivity across the landscape as habitat is lost. The clustered habitat scenario (C) leaves isolated habitat 'islands' that may promote endemism over geological timescales. These habitat 'islands' are conceptually analogous to the oceanic islands in MacArthur and Wilson's theory of island biogeography[4]. Endemic species may thus arise naturally on such islands within neutral simulations. Scenario C directly tests the mechanistic assumption of Sahney et al.[3] (that endemism, driven by fragmentation and manifesting as increasing beta diversity, is the cause of tetrapod diversity increases post-CRC).

Our models of random habitat loss (scenario B) demonstrate that increasing the amount of habitat loss, while keeping all other parameters the same, causes 'global' species richness and beta diversity to decline (Fig. 3 and Extended Data Fig. 1). Species richness decreases relatively linearly across all time periods. However, alpha and beta

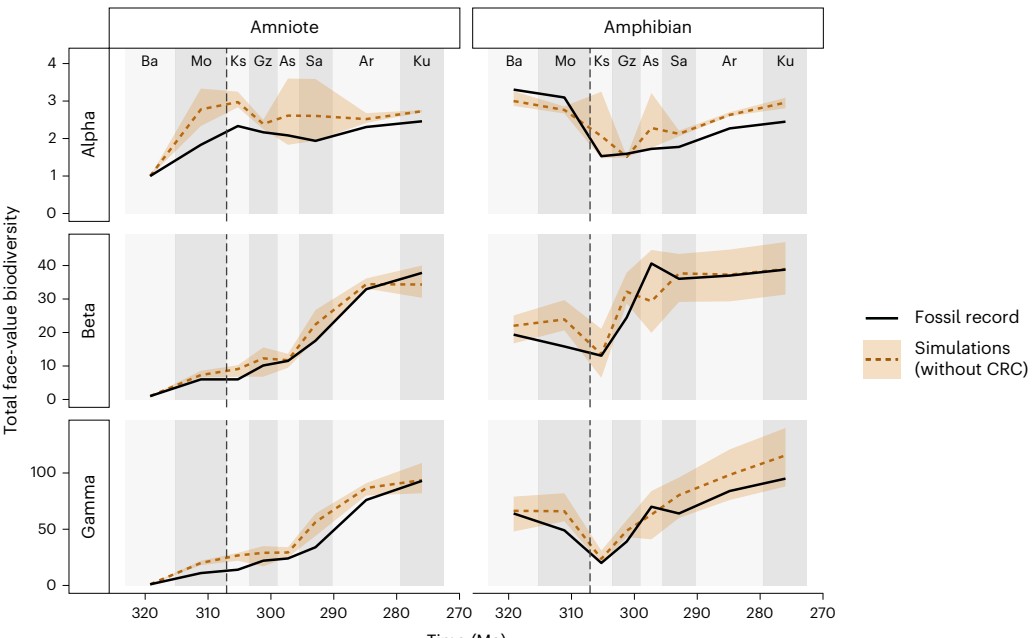

**Fig. 2 | 'Pristine' landscape (scenario A).** Simulated tetrapod diversity patterns over time compared against the raw fossil record. Simulations were performed on a 'pristine' landscape with no habitat fragmentation (that is, the CRC was absent). Three metrics of biodiversity (alpha, beta and gamma diversity; Table 1) are shown for both amphibians and amniotes from Bashkirian to Kungurian from empirical data (solid black lines) and from simulated communities (dashed lines). The shaded area surrounding the dashed lines represents the variation in the 5 best-fitting simulations from a total of 25. The dashed vertical line indicates the timing of the CRC. Interval abbreviations are as in Fig. 1.

diversity demonstrate a more variable pattern across time for different levels of habitat loss. In particular, the interval between 307 and 297 Ma has very similar alpha diversity for all levels of habitat loss. This is potentially caused by the lower numbers of sampled fossils found at this time (Extended Data Fig. 2), because under poor sampling, inferred alpha diversity will be impacted primarily by the number of sampled individuals, rather than by other factors such as the quantity of surrounding habitat.

Our clustered habitat (scenario C) tested whether neutral theory supports the hypothesis that habitat loss results in highly disconnected habitat islands that promote endemism. Under these circumstances, unless the fossil localities were close, dispersal between distinct fossil localities was restricted almost entirely, meaning that the number of shared species between localities was likely to be very low. The neutral simulations of the clustered habitat scenario generated diversity patterns that did not closely fit the empirical fossil data (Fig. 4). Although the overall trend matches to some extent, the simulations had a high level of variability between intervals, primarily dictated by the number of fossil localities known for each interval. Furthermore, loss of habitat, and the resulting decrease in the size of the metacommunity supplying individuals to the fossil sites, caused a reduction in species richness. Similarly, there was also a reduction in alpha diversity, particularly for amniotes.

By simulating this same best-fitting scenario (20% random habitat loss for amniotes and a pristine landscape for amphibians), but sampling more individuals at each locality, it is possible to predict the broader diversity changes under the same model beyond the empirical sample size. When ten times more individuals are sampled from each fossil locality, differences emerge compared with simulations in which sampling of the fossil record is exactly matched (Fig. 5). The general trends in species richness over time for both amniotes and amphibians are roughly similar to the trends observed in the fossil record (Fig. 5). However, there is no longer a significant increase in beta diversity post-CRC, especially for amphibians. Likewise, alpha diversity is relatively consistent over time. There is also a broader range in the simulation outcomes where many more individuals are sampled.

To remove temporal variation in sampling intensity (but retain spatial sampling structure), we also simulated a model version with constant sampling effort over time. When 100 individuals are randomly selected from each time slice, in the same spatial arrangement as the empirically sampled localities, the trend in species richness (gamma diversity) over time tracks the changes in global diversity (Fig. 6). The simulated patterns in diversity where sampling effort is standardized bears only limited resemblance to the real fossil record together with its sampling biases; it matches the general trend only for beta diversity.

## Discussion

This work shows that the apparent increases in face-value diversity observed in the fossil record of early tetrapods across the late Carboniferous/early Permian can be explained by a simple mechanistic neutral model that accounts for biases in sampling. However, there does appear to be a small but observable change in the characteristics of early tetrapod diversity around 307 Ma, the approximate timing of the CRC. This can be explained by either changes in dispersal, changes in density of individual organisms (Extended Data Figs. 3 and 4) or fragmentation of habitats (which is theoretically similar to a reduction in species diversity; Fig. 3). These findings support the previous assessment that patterns of diversity in the early tetrapod fossil record should not be interpreted at face value[6].

The model scenario of rainforest fragmentation that is most consistent with the empirical (face-value) fossil data is one in which the global density of individual early tetrapod organisms decreases by a small amount at 307 Ma (Fig. 3). When sampling the simulations in a realistic manner, this results in a temporary dip in the face-value gamma diversity and beta diversity of amphibians around the time of the CRC. By contrast, amniotes show an increase in both face-value beta and gamma diversity, suggesting a potential role of endemism, although this does not have much effect until 10 million years after the CRC. Under this scenario, simulated face-value gamma diversity losses during the CRC are even greater than those observed at face value in the fossil record (after accounting for the changes in sampling effort over time).

## Table 1 | Glossary of terms used in this study

| Term | Definition |
|---|---|
| Neutral theory | Related to the study of neutral models, these are individual-based models where the fate of individuals (chances of survival, movement and reproduction) is unconnected with species identity. |
| Alpha diversity | The richness (number of taxonomic groups) at different sites or habitats within a local scale. Also referred to as 'local richness'. In this study, we look at species richness. Alpha diversity at a given time point is calculated here as the number of species per locality, as a mean across all localities at that time point. |
| Gamma diversity | The total diversity (number of taxonomic groups) across all communities within a larger region. Also referred to as 'global diversity'. In this study, we consider species-level diversity and calculate gamma diversity as the total number of species across all localities at a given time point. |
| Beta diversity | The ratio between alpha (local) and gamma (global) diversity given by the equation $\beta = \frac{\gamma}{\alpha}$. Beta diversity quantifies the difference between communities in the region. Increased beta diversity means increased turnover between localities. Beta diversity can sometimes be defined differently from this, but it always aims to capture turnover of diversity. |
| Face value | Face-value (or 'raw' or 'uncorrected') alpha, beta or gamma diversity is the measured diversity seen in a possibly biased sample. By contrast, the true alpha, beta or gamma diversity corresponds to what is really present or what is seen from an unbiased sample. |
| Amniote | Tetrapod species belonging to the clade Reptiliomorpha, which includes the crown group Amniota and those species more closely related to them than to modern amphibians. |
| Amphibian | Non-amniote tetrapod species including early Tetrapodpodomorpha, non-amniote tetrapodomorph species including Lepospondyli and Temnospondyli. |

When many more individuals were sampled from the same simulation models, the emergent diversity patterns changed considerably because so much more of the underlying system is revealed (Fig. 5). For example, a larger sample may not uncover much more species-level diversity, suggesting that the already present species dominate with large abundances. This sensitivity to sampling suggests that the temporal changes in alpha and beta diversity found in the fossil record may disappear as more fossils are found. This shows that the effect of sampling bias can be mitigated to an extent by more intense sampling, even if the additional sampling is equally biased. When the same number of individuals are sampled from each point in time within our simulations, the trends in species richness and alpha diversity disappear to an extent (Fig. 6). This, again, suggests that the face-value patterns in the fossil record are an artefact of changes in the number of locations sampled within each time interval. The development of endemism does happen, as can be seen from increasing beta diversity following the CRC, for both amniotes and amphibians (Fig. 6). However, it is not enough to offset the alpha diversity decrease from habitat loss, suggesting that the effects of endemism often do not increase gamma diversity[22] and in fact there is a small decrease in gamma diversity after the CRC, probably in response to habitat loss.

Taken together, our results suggest that endemism from habitat loss at the CRC would have probably led to a net decrease in gamma species richness, and not an increase as has been claimed previously by Sahney et al.[3]. After accounting for sampling bias, the limited changes to global richness are primarily driven by a modest reduction in global tetrapod population density over time, which is consistent with the expected ecological impact of the collapse of the rainforests and drying of the climate. The simulated scenario that aligns best with the empirical, face-value patterns is that of random habitat loss of between 0% and 20%, a scenario that is dynamically identical under neutral theory to an equivalent reduction in density[17].

Our models used relatively abstract patterns of habitat loss, because the real patterns are not known. Future research could attempt to produce more realistic patterns of rainforest habitat loss, based on either palaeoclimate reconstructions or comprehensive occurrence data for fossil plants. Integrating more accurate maps of tropical rainforest coverage over time with the mechanistic basis of neutral theory would be more informative for exploring theories of diversity generation following the CRC. This is not currently possible because of the absence of readily available palaeoclimate reconstructions for this particular time interval, and the challenges associated with building comprehensive, spatially explicit, occurrence-based dataset for fossil plants. It is not immediately clear how one would relate forest patterns to the dynamics of early tetrapod diversity because amphibians and reptiles (both modern and extinct) exhibit broad variability in their dependency on forest cover. One immediate pattern of rainforest loss that could be incorporated into future related work, with the addition of empirical data, is the hypothesis that the rainforest disappearance began in western Pangaea before moving eastwards[23]. Another key consideration for future research is deciphering the influence of hierarchical spatial scaling on the patterns recovered here; alpha, beta, and gamma diversity are ultimately nested and changes at the community scale can be reflected at larger scales[24,25].

The neutral models explored here assumed that abundances (population densities) were consistent over time (that is, the same number of individual organisms exist in each unit of habitat), except in the case of habitat loss at the CRC. However, the abundance of early tetrapods would also have a significant effect on the numbers of specimens preserved in the fossil record. Consequently, lower numbers of fossil specimens could be indicative of smaller populations and lower species richness. However, it is difficult to satisfactorily resolve the relationship between density and sampling rate because the nature of fossil preservation varies substantially over time, space and environments. Across our dataset of late Carboniferous and early Permian tetrapods, quality of preservation (and thus the size of the 'taphonomic window') varies substantially, which in turn influences sampling intensity (Extended Data Fig. 5). Fossil localities of late Carboniferous age that have yielded particularly well-preserved or abundant specimens are typically coal deposits[26,27] (for example, coal mines at Nyrǎny in the Czech Republic and Linton Diamond Mine in Ohio, United States). In the early Permian, owing to the combination of orogenic activity (mountain building) and drier climatic conditions, fossils are much less likely to be preserved in coal deposits. Instead, many richly diverse localities in the early Permian are the remains of terrestrial environments such as floodplains, river systems and even caves[28], many of which have been quarried and excavated extensively over many decades (for example, various localities in the Red Beds of Texas and Oklahoma, United States). This lack of coal deposits in the early Permian also reflects the contraction of rainforest habitats across this interval, invoking the common-cause hypothesis, which states that the covariation of fossil and rock records is due to an external factor[29,30]. Similarly, the disappearance of coal deposits might simultaneously affect taphonomic windows and true underlying biodiversity driven by the loss of rainforest habitat. Because of these temporal changes in preservation, it is impossible to infer true densities of early tetrapods during this interval (and probably any interval in the geological past). This limitation motivated keeping density as a free parameter within the neutral simulations, but precludes understanding of how both early tetrapod densities and preservation rates varied. Unravelling the true historical changes would require a better understanding of both the true densities of early tetrapods over time and changes to the preservation rates over time (one of the measures that is possible to estimate for species within assemblages).

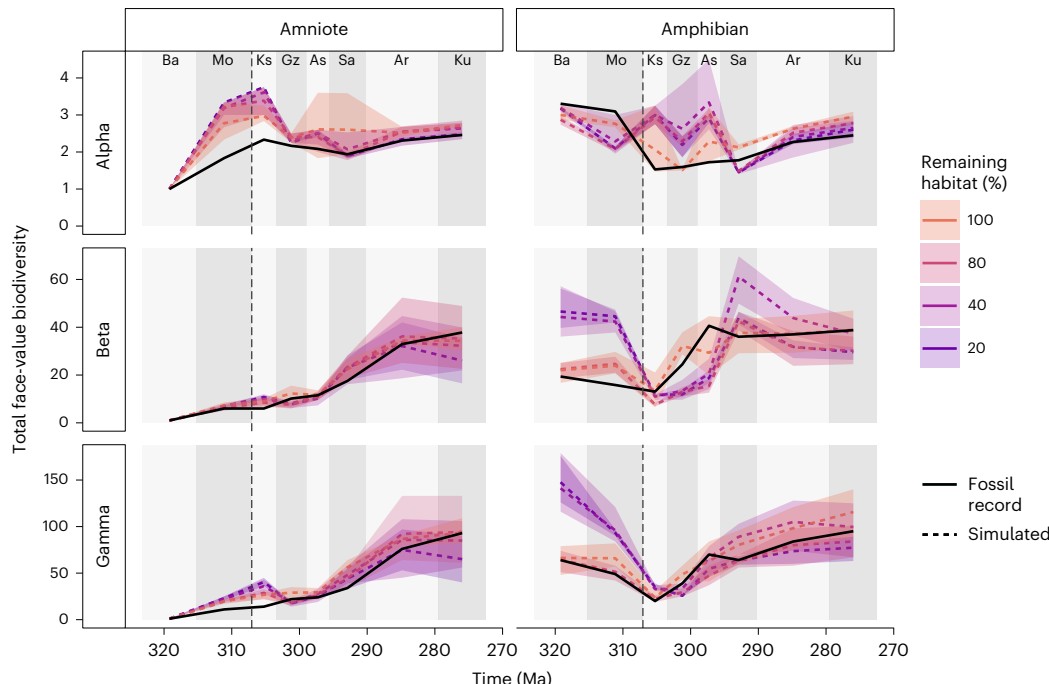

**Fig. 3 | Random habitat loss (scenario B).** Simulated tetrapod diversity patterns over time compared against the fossil record (face-value, unstandardized counts of species). Simulations were performed on a landscape where the impact of the CRC is represented by random habitat loss occurring at 307 Ma (dashed line). Predictions of biodiversity patterns are produced by a neutral model parameterized with a percentage of habitat remaining (for example, 80% habitat remaining is equal to 20% loss). The shaded areas surrounding the dashed lines represent the variation in the five best-fitting simulations. The dashed vertical line at 307 Ma indicates the timing of the CRC. For definitions of diversity measures see Table 1. Interval abbreviations are as in Fig. 1.

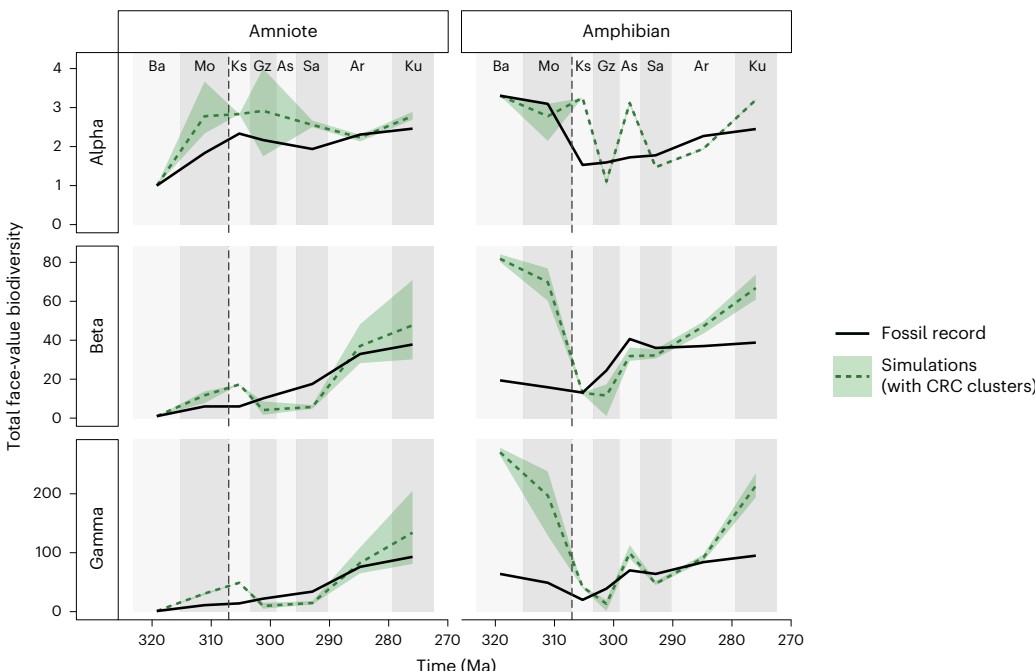

**Fig. 4 | Clustered habitat scenario (scenario C).** Simulated tetrapod diversity over time from neutral models compared against the fossil record (face-value, unstandardized counts of species). Simulations were performed on a landscape where the impact of the CRC is represented by clustered habitat loss occurring at 307 Ma (dashed line). The remaining habitat following the CRC is associated into habitat islands 100 km in diameter at the locations of each sampled fossil locality. The shaded area surrounding the dashed lines represents the variation in the five best-fitting simulations. The shaded areas represent the variation in the five best-fitting simulations. The dashed vertical line at 307 Ma indicates the timing of the CRC. For definitions of diversity measures see Table 1. Interval abbreviations are as in Fig. 1.

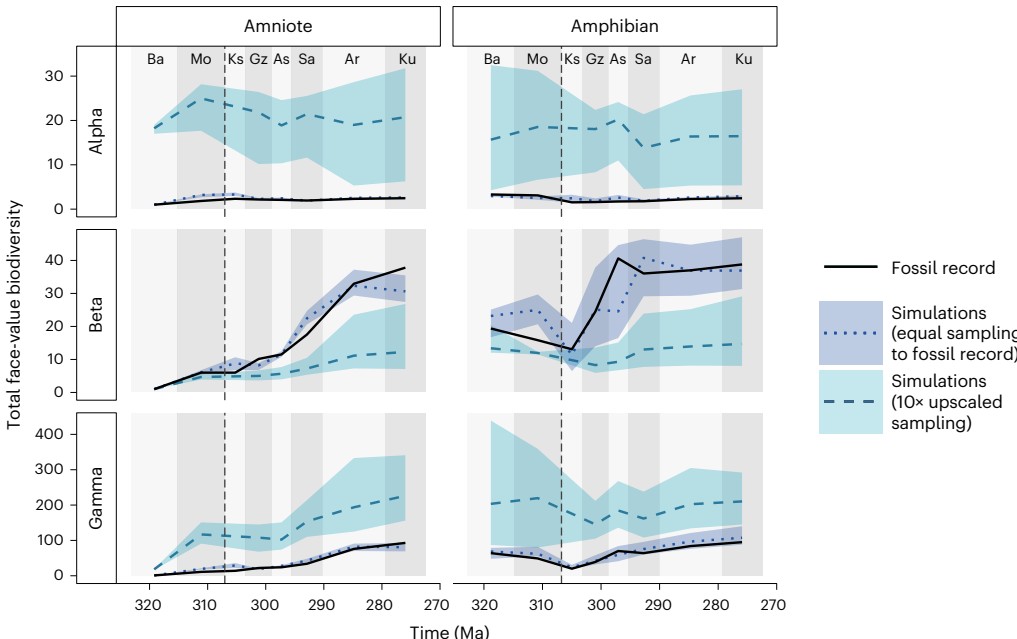

**Fig. 5 | 'Upscaled diversity' from the fossil record using neutral models.**
The mean values from simulations of the five best-fitting parameters from the scenario under random habitat loss are shown by the dark blue dotted lines. The grey dashed lines represent the same simulations, but sampling ten times more individuals than are present in the fossil record. The shaded areas surrounding the dashed lines represent the variation in the five best-fitting simulations. The dashed vertical line at 307 Ma indicates the timing of the CRC. For definitions of diversity measures see Table 1. Interval abbreviations are as in Fig. 1.

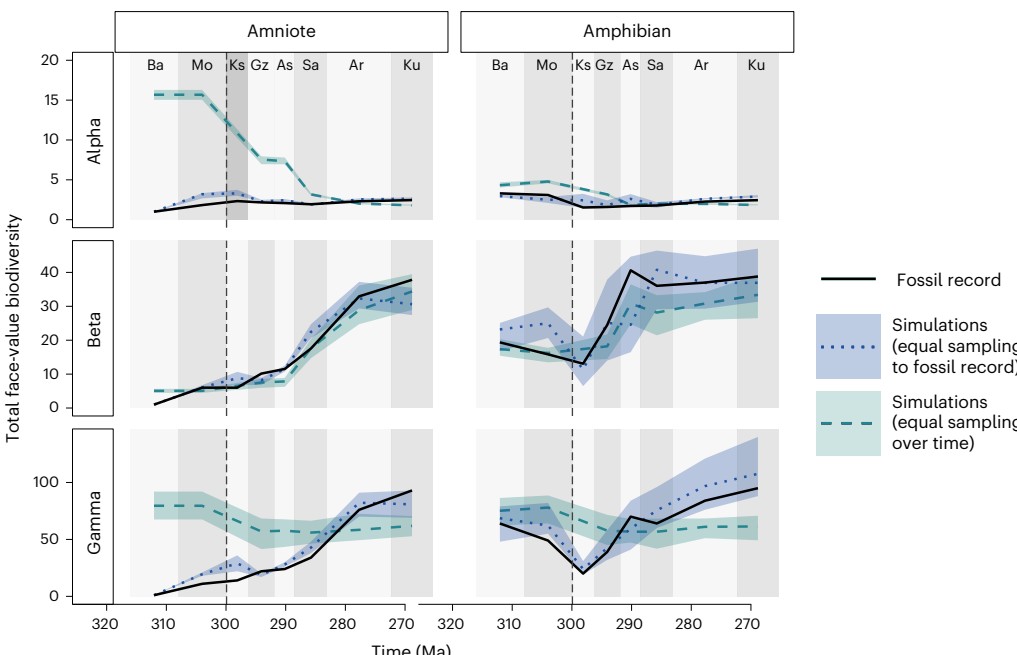

**Fig. 6 | Simulated diversity where temporal biases are removed but spatial sampling structure is retained.** Solid black lines represent the empirical data from the fossil record. The grey dashed lines represent the same simulations, but randomly sampling 100 individuals for each interval. These simulations are without any temporal bias in sampling frequency, in contrast to the fossil record and preceding simulations where the number of individuals sampled changes over time. The shaded areas represent the variation in the five best-fitting simulations. The dashed vertical line at 307 Ma indicates the timing of the CRC. For definitions of diversity measures see Table 1. Interval abbreviations are as in Fig. 1.

Our explanations of the changes in early tetrapod diversity through time have all been based on ecological neutral theory. Alternative explanations could come from changes in non-neutral dynamics, such as species niche structure, competition between species or wider ecosystem-level shifts. These explanations cannot yet be tested from a mechanistic basis but represent an exciting avenue of future research, as do investigations of the minimal requirements for these models to have stronger predictive power.

## Conclusions

Statistical approaches to estimating past biodiversity patterns can provide important insights into patterns of diversity[6,22,31]. However, they are generally limited by the geographical and temporal extent of the available fossil occurrence data. In our study, spatially explicit neutral models have proven to be a valuable tool for directly testing established hypotheses of diversity change in the first vertebrates to emerge onto land, and illuminating the impacts of spatial and temporal sampling biases on their face-value diversity patterns.

Interdisciplinary studies integrating modern ecological theory with palaeontological data have been identified as crucial for informing predictions for future diversity[32,33] as well as more accurately understanding past biodiversity patterns[34–37]. Our results shed new light on the impact of the CRC on early tetrapod diversity, by showing that increased endemism resulting from habitat loss at the CRC is unlikely to have produced an increase in biodiversity. Our study also offers new insights into the effects of sampling bias on fossil diversity estimates, and demonstrates the huge untapped potential that mechanistic models, such as those founded on neutral theory, have for testing hypotheses of deep-time biodiversity change.

## Methods

### Neutral models

Assessment of spatial and temporal biases on a mechanistic basis requires a model that is spatially and temporally explicit. In addition, to study the impact of habitat loss and fragmentation on biodiversity requires a model that can directly incorporate these dynamics within the biodiversity-generating process. Neutral models fulfil all these requirements and are also tractable at large scales. Neutral theory[38] assumes that the properties of an individual are independent of its species identity. The dynamics of neutral models are thus dictated by some combination of dispersal, ecological drift and speciation. The output of neutral models is a simulated ecological community, where each individual has an assigned species identity. These simulated communities are equivalent to a complete census of the simulated area. The communities provide a baseline for expected biodiversity under 'idealized' conditions[39], against which the biodiversity from real communities can be compared. Neutral theory has, however, only rarely been applied in analyses of fossil data. A few palaeoecological studies have used spatially implicit neutral theory[16,20,21] where populations (for example, within separate continents) are divided to roughly represent spatial barriers. To the best of our knowledge, however, no previous study has applied a fully spatially explicit neutral model to fossil data.

The classic, spatially implicit model[38] conceives of a local community connected to a metacommunity by immigration at a given rate; other models incorporate more explicit dispersal between parts of the landscape. Being based on fundamental biological mechanisms, neutral theory has utility for identifying underlying dynamics[40], acting as a null or 'ideal' model[39], or making predictions at broader spatial or temporal scales than are possible with field experiments[41]. We use a spatially explicit neutral model[42] that incorporates the exact locations of each individual in space and incorporates a dispersal kernel to describe the distance moved by offspring from their parents. Such a fully spatially explicit model is essential to account for spatial sampling bias. The metacommunity concept of the spatially implicit model is replaced by movement around a broad spatially explicit landscape.

The mechanism of our model proceeds as follows: an individual is first chosen to die leaving a 'space' that will be filled by a newborn individual. The parent of the newborn individual is chosen from other nearby individuals according to a dispersal kernel, which we modelled as a two-dimensional normal distribution. The newborn is normally conspecific to its parent, but occasionally, with probability $v$ at each birth, it becomes a new species. Over many generations, nearby individuals are more likely to be the same species, whereas distant individuals will be more likely to be different. We use these models to generate communities of species across the landscape.

A major development for neutral theory was backwards-time coalescence methods[43], which produce equivalent results to a naïve (forwards-time) implementation of the mechanisms described above but are many orders of magnitude faster in computational performance. Furthermore, many scenarios are made possible with coalescence that are not possible otherwise, such as exceedingly large or infinite landscapes[42] or sampling a small subset of individuals from the landscape without having to simulate the entire landscape first. The latter feature means that our models can simulate observations at just the precise locations observed in the fossil record, while accounting mechanistically for the whole community alive at the time with a full spatial structure from the relevant period in history. An equivalent model using forwards-time techniques would require simulating every tetrapod that existed across the entire time frame and continent of interest, a feat not remotely feasible with current computational power. Unfortunately, most non-neutral models cannot benefit from the use of coalescence and associated abilities to account for sampling in huge spatially explicit systems. We use the pycoalescence package available for Python and R[17], which uses coalescence methods implemented in C++ for high-performance spatially explicit neutral simulations.

### Preparation of fossil occurrence data

Data detailing the global occurrences of early tetrapod species from the late Carboniferous (Bashkirian) to early Permian (Kungurian) were downloaded from the Paleobiology Database (www.paleobiodb.org). These data represent the published knowledge on the global occurrences of early tetrapod species alongside taxonomic opinions; it is the result of a concerted effort to document the Palaeozoic terrestrial tetrapod fossil record. The dataset was cleaned by removing marine taxa, ichnotaxa and taxa with uncertain taxonomic identifications. The total number of amniote (including Reptiliomorpha (Table 1)) and amphibian (non-amniotes and early tetrapodomorphs (Table 1)) species per locality was ascertained and recorded (Extended Data Fig. 5). The resulting dataset (Supplementary information) details the number of amniote and amphibian species found at each locality (a 'collection' in Paleobiology Database terms) during each of the eight stratigraphic intervals from the Bashkirian to the Kungurian.

### Neutral simulations of early tetrapods

We split the tetrapods into amphibians and amniotes to reflect their differing physiologies and environmental preferences, treating each with an independent neutral model. Our simulations required maps of the relative density of individuals across the globe. These were determined separately for each interval (Bashkirian to Kungurian) from the continental boundaries of the time. Global rasterized maps of individual relative densities were produced at 0.01-degree resolution using the continental extents provided by the Paleobiology Database based on GPlates palaeogeographical reconstructions[44]. This corresponds to pixels of around 1 km² each representing a cell for our model. The palaeocoordinates of each fossil locality were calculated and localities were then aggregated within each 1 km² cell. Specimen counts per locality were estimated using the 'occurrences-squared' heuristic[45], calculated simply as the square of the number of unique fossil occurrences. This metric provides a basic way of accounting for the fact that most localities lack information about counts of specimens, and because it is rarely obvious how many distinct individuals contributed to a set of fossil fragments. Using this metric in our models approximates the total number of individuals that contributed to the observed fossil record and therefore the number of individuals that should be sampled in the neutral simulations. This generated a 'sample map' defining the number of individuals to be sampled at each position in space. Because the majority of the globe was not sampled, most cells in this sample map were set to 0. The relative density and

the sample maps together contain the spatial information of the entire global community of amphibians and amniotes for the simulation and define which individuals from each global community were sampled.

The second parameter critical for the simulations is the dispersal rate ($\sigma$), which controls the distance that individuals disperse across the landscape in a given generation. $\sigma$ is used as the variance in a Rayleigh distribution determining the radius of dispersal, with a separate uniform random number determining the direction of dispersal. This means that larger values of $\sigma$ correspond with longer dispersal distances, on average.

The eight stratigraphic intervals sampled from the fossil record were sufficiently far apart in time that we reasonably assumed no shared species between the different time intervals within the model. Consequently, we ran simulations for each time interval as separate neutral models in parallel, and aggregated the communities post-simulation.

We performed simulations with parameters encompassing a broad range of biologically feasible values: density values for habitat cells ranged from 25 to 1,000 individuals per km for 'habitat' regions (non-habitat regions have a density of 0 individuals), the parameter of dispersal ($\sigma$) varied to give mean distances of 0.1–14 km, and speciation rates varied from $10^{-8}$ to $10^{-1}$. We explored 5 density and 5 dispersal parameters giving 25 combinations using Latin hypercube sampling[46] to evenly sample from arithmetic parameter space. Under coalescence methods, higher speciation rates can be applied post-simulation for generating communities[17,43]. We performed simulations using a minimum speciation rate of $10^{-8}$ and applied all other speciation rates afterwards to generate additional communities.

Three broad scenarios of tetrapod diversity were simulated (Fig. 1). In all models, the global landscape was restricted by continental boundaries. Our simplest model (scenario A) contained pristine habitat with no habitat loss (that is, uniform, with no habitat fragmentation) Two scenarios (B and C) exhibited habitat loss of different forms following the CRC. The landscape was fragmented according to a random spatial pattern, so that land areas contained habitat on a percentage of their area (either 20%, 40% or 80% of habitat remaining). The random pattern was generated by randomly removing pixels from the landscape until the desired percentage of habitat remains.

## Model parameterization

To determine how well the simulations fit the patterns in the fossil record, four biodiversity metrics were used for each interval: the alpha diversity ($\alpha$) for each fossil locality (that is, the local species richness), the mean alpha diversity across all localities, the total species richness across all localities ($\gamma$) and the mean beta diversity (calculated as $\beta = \frac{\gamma}{\alpha}$) across all localities. The mean actual percentage error between the real and simulated fossil records in alpha diversity for each locality was averaged to get a mean alpha accuracy $\mu_\alpha$. The mean actual percentage error between the real and simulated fossil records was calculated for each other metric ($\alpha$, $\beta$ and $\gamma$). Averaging the mean actual percentage errors for the four metrics ($\mu_\alpha$, $\alpha$, $\beta$ and $\gamma$) gives an indication of the goodness of fit for one simulation—we refer to this percentage as the accuracy of a single simulation. There is some redundancy between the values because the parameters are not independent, but the approach should still result in the simulation that most closely matches the real fossil record.

Because each interval was run as a separate neutral simulation, the parameters of speciation rate, density and dispersal could be allowed to vary over time. However, because combinations of parameters can be aggregated in any number of ways, we considered just two possibilities that reflected our assumptions of the possible ecological changes over time: either there was no change in these parameters (we use a single parameter set for all intervals); or the parameters could change at the time of the CRC (we use two parameter sets, one for pre-CRC (323–307 Ma) and one for post-CRC (307–372 Ma)). The first scenario represents a neutral ecosystem with no changes in fundamental ecological dynamics. The second presents a neutral scenario that assumes ecological changes were generated by the CRC and may be reflected in neutral dynamics. In some tests a single set of parameters (speciation rate, dispersal and density) was used for all time intervals, whereas in others this requirement was relaxed to investigate how the parameters themselves may change over time.

## Upscaling and downscaling simulated communities

To explore potential biodiversity patterns that would emerge if the fossil record included a larger number of individuals, we ran simulations with the same model parameters as the best-fitting simulations, including the same number of simulated individuals, but reporting back on ten times more sampled individuals (sampled with replacement). This scenario demonstrates how the emergent biodiversity patterns change with the overall intensity of sampling effort in isolation from other factors. We also explored the effect of sampling a fixed number of individuals from each time interval, for comparison with sampling different numbers of individuals from each time interval in line with the temporal changes in sampling intensity present in the fossil record. This samples from the simulation without sampling-intensity biases over time but retains the sampling-intensity biases over space matching the real-world spatial sampling pattern. It enables us to demonstrate the effect of temporal sampling biases in isolation from other factors.

## Reporting summary

Further information on research design is available in the Nature Portfolio Reporting Summary linked to this article.

## Data availability

All relevant data supporting our analyses are available in the OSF repository: https://doi.org/10.17605/OSF.IO/ZGHWB.

## Code availability

Code for downloading and cleaning fossil occurrence data from the Paleobiology Database is available at: https://github.com/emmadunne/neutral_theory_tetrapods. Code for running simulations using *pycoalescence* is available at: https://github.com/thompsonsed/palaeo_neutral_sims. Both are accessible through the OSF repository: https://doi.org/10.17605/OSF.IO/ZGHWB

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

## Acknowledgements

We thank all contributors to the Paleobiology Database, in particular T. Liebrecht, R. Whatley, J. Dummasch, J. Alroy and M. Carrano. E.M.D. thanks T. Dunkley-Jones and P. Mannion for helpful comments and discussion. E.M.D., R.A.C. and R.J.B. were funded by the European Union's Horizon 2020 research and innovation programme under grant agreement 637483 (European Research Council Starting Grant TERRA to R.J.B.), and also through a Leverhulme Research Project Grant (RPG-2019-365 to R.J.B.). R.A.C. was also funded by a Royal Society University Research Fellowship (RF\ERE\210396). S.E.D.T. was funded by the Imperial–National University of Singapore Joint PhD Scholarship. J.R. was funded by a Natural Environment Research Council fellowship (NE/L011611/1) and a Leverhulme Trust Research Fellowship (RF-2022-497). Through J.R. and S.E.D.T., this study is an output of the Georgina Mace Centre for the Living Planet at Imperial College London. All simulations were performed on high-throughput computing systems at Imperial

College London. For the purpose of open access, the authors have applied a 'Creative Commons Attribution' (CC BY) licence to any author accepted manuscript version arising. This is Paleobiology Database official publication number 454.

## Author contributions

R.A.C. and J.R. conceived the project and all authors input into the design. E.M.D. and S.E.D.T. curated the data, conducted the analyses, prepared the figures and led the writing of the paper. All authors contributed to collating material for the supplementary information, and to the writing and approval of the final paper.

## Competing interests

The authors declare no competing interests.

## Additional information

**Extended data** is available for this paper at https://doi.org/10.1038/s41559-023-02128-3.

**Correspondence and requests for materials** should be addressed to Emma M. Dunne.

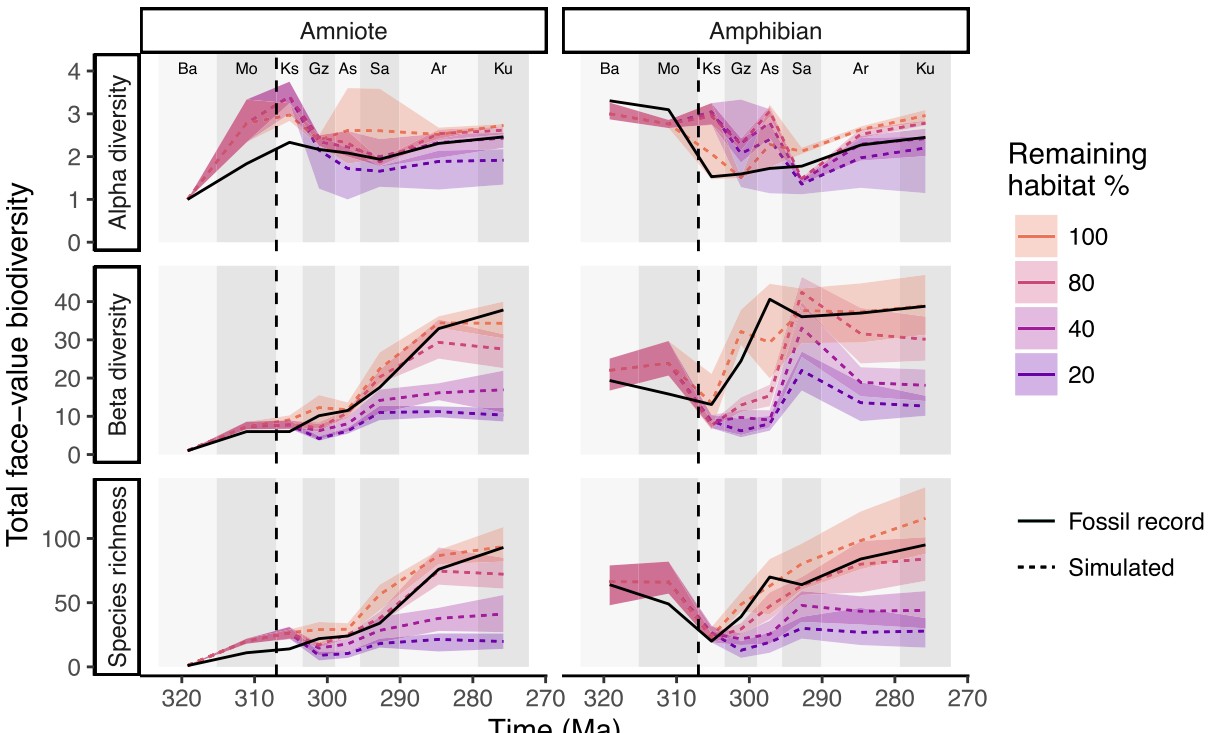

**Extended Data Fig. 1 | Simulated tetrapod diversity patterns over time compared against the fossil record, including habitat loss.** Simulated tetrapod diversity patterns over time compared against the fossil record (that is face-value, unstandardized counts of species). Predictions of tetrapod biodiversity patterns are produced by a neutral model parameterised with 80% habitat remaining (20% loss) and then simulated with different levels of remaining habitat (that is 100%, 40% and 20% habitat remaining). The shaded areas surrounding the dashed lines represent the variation in the five best fitting simulations. The dashed vertical line at 307 Ma indicates the timing of the CRC. For definitions of diversity measures see Table 1. The following abbreviations are used for intervals: 'Ba' = Bashkirian, 'Mo' = Moscovian, 'Ks' = Kasimovian, 'Gz' = Gzhelian, 'As' = Asselian, 'Sa' = Sakmarian, 'Ar' = Artinskian and 'Ku' = Kungurian.

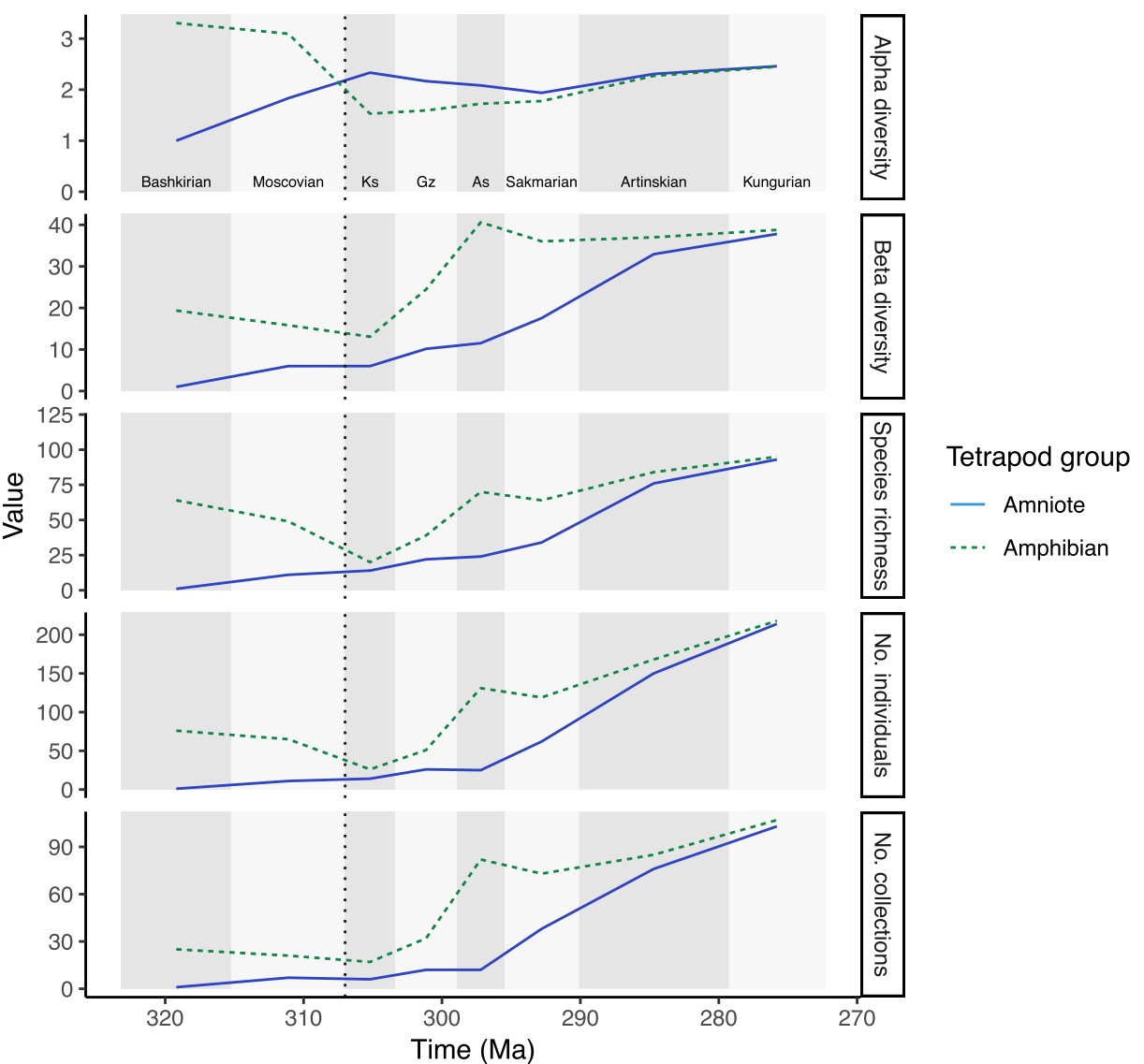

**Extended Data Fig. 2 | Biodiversity metrics through time using uncorrected fossil data.** Raw data from the fossil record, indicating biodiversity metrics (alpha diversity, beta diversity and total species richness) and the number of individuals (that is fossils) and collections over time. Interval abbreviations are as in Extended Data Fig. 1.

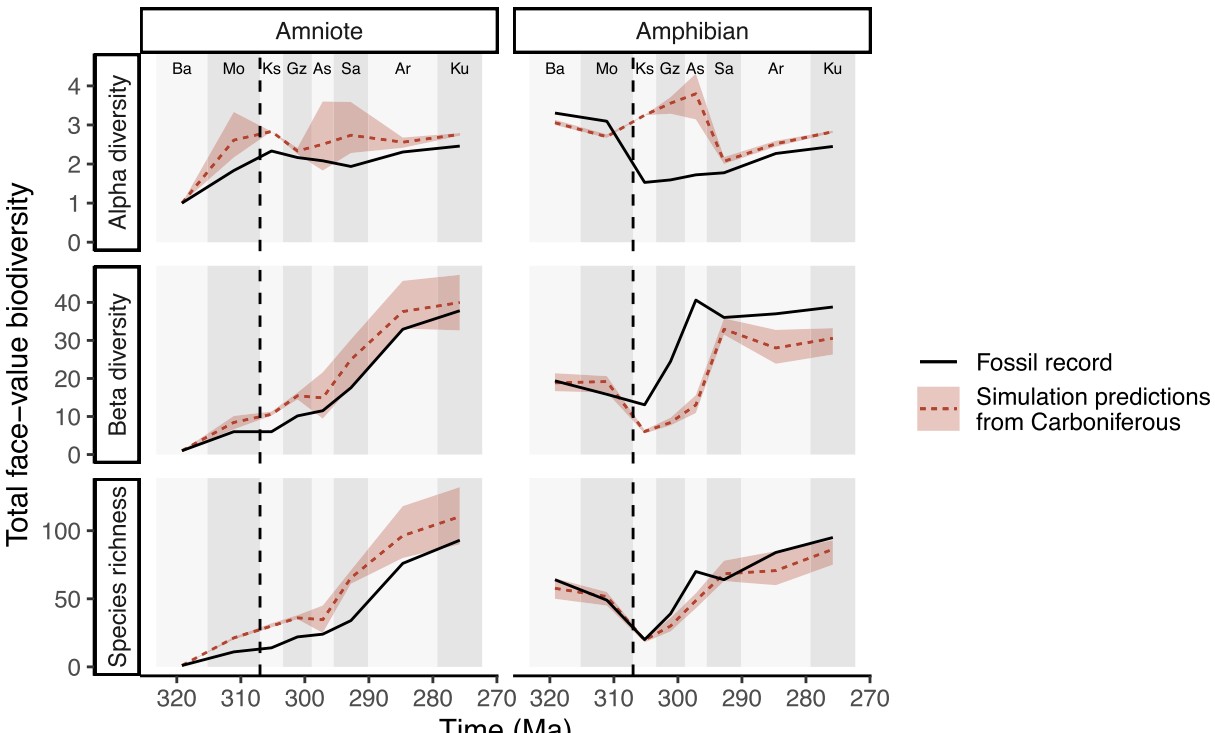

**Extended Data Fig. 3 | Predictions of diversity from neutral model parameterised on Carboniferous diversity only.** Simulated tetrapod diversity patterns over time compared against the fossil record (that is face-value, unstandardized counts of species). Predictions of tetrapod diversity from a neutral model parameterised solely on Carboniferous diversity. Three metrics of biodiversity (alpha, beta, and gamma diversity; Table 1) are shown for both amphibians and amniotes from the Bashkirian to Kungurian from empirical data (solid black lines) and from simulated communities (dashed lines). The shaded areas surrounding the dashed lines represent the variation in the five best fitting simulations. The dashed vertical line at 307 Ma indicates the timing of the CRC. For definitions of diversity measures see Table 1. Interval abbreviations are as in Extended Data Fig. 1.

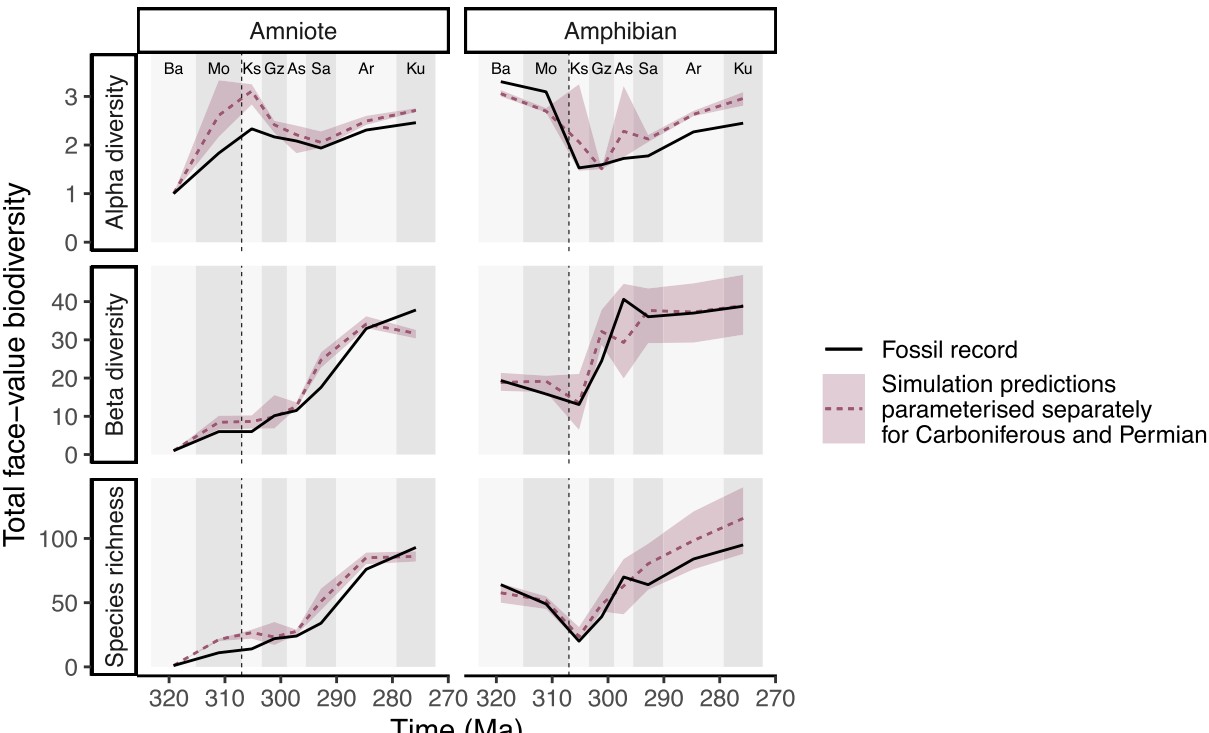

**Extended Data Fig. 4 | Predictions of diversity from neutral model parameterised on both Carboniferous and Permian diversity.** Simulated tetrapod diversity patterns over time compared against the fossil record (that is face-value, unstandardized counts of species). Predictions of tetrapod biodiversity patterns are produced by a neutral model parameterised separately for the late Carboniferous (pre-307 Ma) and Permian (post-307 Ma). The shaded areas surrounding the dashed lines represent the variation in the five best fitting simulations. The dashed vertical line at 307 Ma indicates the timing of the CRC. For definitions of diversity measures see Table 1 (main text). Interval abbreviations are as in Extended Data Fig. 1.

### Bashkirian: 323.2–315.2 Ma

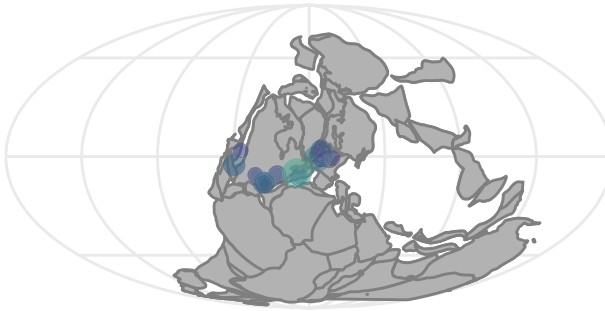

### Moscovian: 315.2–307 Ma

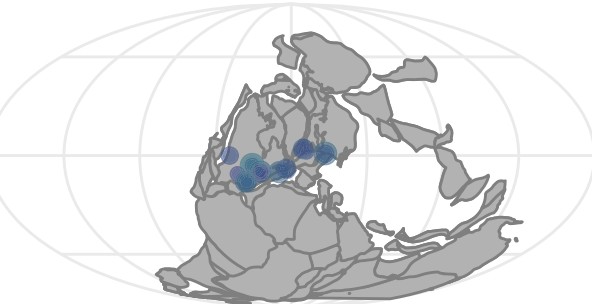

### Kasimovian: 307–303.4 Ma

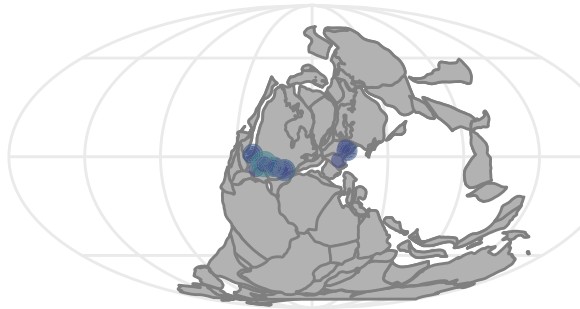

### Gzhelian: 303.4–298.9 Ma

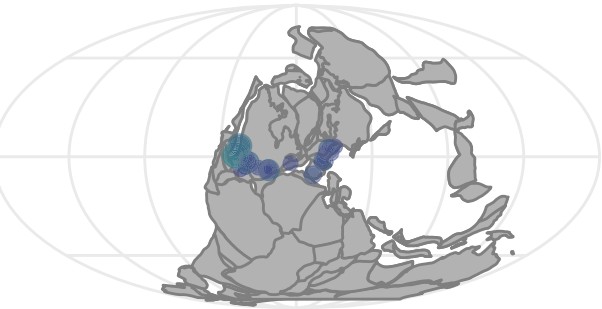

### Asselian: 298.9–295.5 Ma

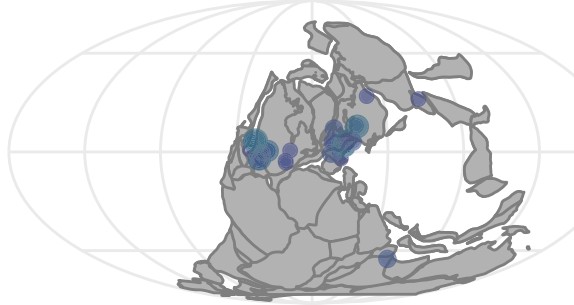

### Sakmarian: 295.5–290.1 Ma

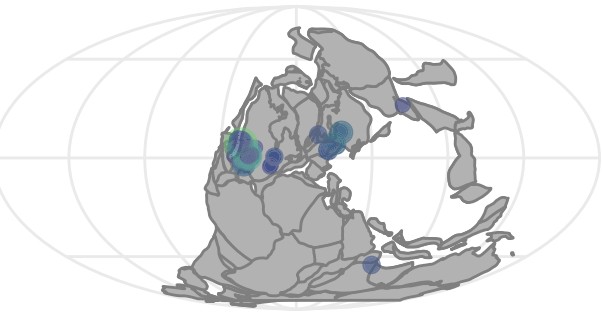

### Artinskian: 290.1–279.3 Ma

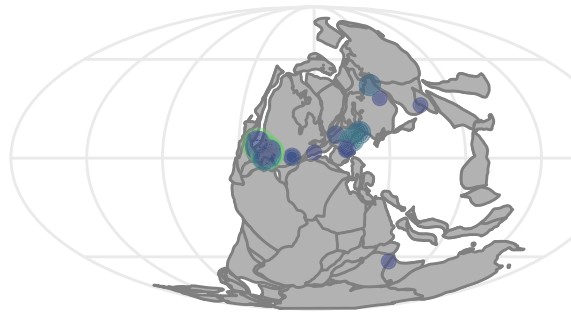

### Kungurian: 279.3–272.3 Ma

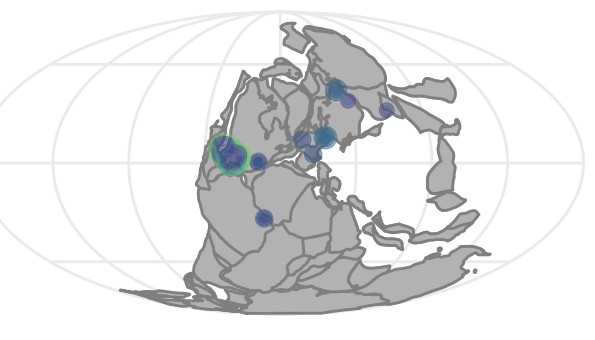

Number of fossils · 0 ● 5 ● 10 ● 15 ● 20

**Extended Data Fig. 5 | Palaeogeographical maps of fossil localities in each study interval.** Global palaeogeographical maps showing the localities of fossil sites in each stage of the late Carboniferous and early Permian. The size and colour of each circle corresponds to the number of species found at each site. Continental configurations are provided by GPlates via the chronosphere R package.

# Reporting Summary

## Statistics

For all statistical analyses, confirm that the following items are present in the figure legend, table legend, main text, or Methods section.

| n/a | Confirmed | |
|---|---|---|
| ☒ | ☐ | The exact sample size (*n*) for each experimental group/condition, given as a discrete number and unit of measurement |
| ☒ | ☐ | A statement on whether measurements were taken from distinct samples or whether the same sample was measured repeatedly |
| ☒ | ☐ | The statistical test(s) used AND whether they are one- or two-sided<br>*Only common tests should be described solely by name; describe more complex techniques in the Methods section.* |
| ☒ | ☐ | A description of all covariates tested |
| ☒ | ☐ | A description of any assumptions or corrections, such as tests of normality and adjustment for multiple comparisons |
| ☒ | ☐ | A full description of the statistical parameters including central tendency (e.g. means) or other basic estimates (e.g. regression coefficient) AND variation (e.g. standard deviation) or associated estimates of uncertainty (e.g. confidence intervals) |
| ☒ | ☐ | For null hypothesis testing, the test statistic (e.g. $F$, $t$, $r$) with confidence intervals, effect sizes, degrees of freedom and $P$ value noted<br>*Give P values as exact values whenever suitable.* |
| ☒ | ☐ | For Bayesian analysis, information on the choice of priors and Markov chain Monte Carlo settings |
| ☒ | ☐ | For hierarchical and complex designs, identification of the appropriate level for tests and full reporting of outcomes |
| ☒ | ☐ | Estimates of effect sizes (e.g. Cohen's *d*, Pearson's *r*), indicating how they were calculated |

*Our web collection on statistics for biologists contains articles on many of the points above.*

## Software and code

Policy information about availability of computer code

| Data collection | Global occurrence data for early tetrapod species from the late Carboniferous (Bashkirian) to early Permian (Kungurian) were downloaded from the Paleobiology Database (www.paleobiodb.org) and are available both through GitHub repository https://github.com/emmadunne/neutral_theory_tetrapods and accessible through in the OSF repository: https://doi.org/10.17605/OSF.IO/ZGHWB |
|---|---|
| Data analysis | pycoalescence package in R (also available for Python) |

For manuscripts utilizing custom algorithms or software that are central to the research but not yet described in published literature, software must be made available to editors and reviewers. We strongly encourage code deposition in a community repository (e.g. GitHub). See the Nature Portfolio guidelines for submitting code & software for further information.

## Data

Policy information about availability of data

All manuscripts must include a data availability statement. This statement should provide the following information, where applicable:

- Accession codes, unique identifiers, or web links for publicly available datasets
- A description of any restrictions on data availability
- For clinical datasets or third party data, please ensure that the statement adheres to our policy

Fossil occurrence data from the Paleobiology Database is available at: https://github.com/emmadunne/neutral_theory_tetrapods and accessible through in the OSF repository: https://doi.org/10.17605/OSF.IO/ZGHWB

# Field-specific reporting

Please select the one below that is the best fit for your research. If you are not sure, read the appropriate sections before making your selection.

☐ Life sciences    ☐ Behavioural & social sciences    ☒ Ecological, evolutionary & environmental sciences

For a reference copy of the document with all sections, see nature.com/documents/nr-reporting-summary-flat.pdf

# Ecological, evolutionary & environmental sciences study design

All studies must disclose on these points even when the disclosure is negative.

| | |
|---|---|
| Study description | This study uses a spatially explicit mechanistic model, based on neutral theory, to test hypotheses of early tetrapod diversity change during the late Carboniferous and early Permian |
| Research sample | Global occurrence data for early tetrapod species during the study interval (late Carboniferous–early Permain) were downloaded from the Paleobiology Database (www.paleobiodb.org) |
| Sampling strategy | All available data were utilised |
| Data collection | N/A |
| Timing and spatial scale | Late Carboniferous–Early Permian (323–272 million years ago) |
| Data exclusions | N/A |
| Reproducibility | N/A |
| Randomization | N/A |
| Blinding | N/A |

Did the study involve field work?    ☐ Yes    ☒ No

# Reporting for specific materials, systems and methods

We require information from authors about some types of materials, experimental systems and methods used in many studies. Here, indicate whether each material, system or method listed is relevant to your study. If you are not sure if a list item applies to your research, read the appropriate section before selecting a response.

## Materials & experimental systems

| n/a | Involved in the study |
|---|---|
| ☒ | ☐ Antibodies |
| ☒ | ☐ Eukaryotic cell lines |
| ☒ | ☐ Palaeontology and archaeology |
| ☒ | ☐ Animals and other organisms |
| ☒ | ☐ Human research participants |
| ☒ | ☐ Clinical data |
| ☒ | ☐ Dual use research of concern |

## Methods

| n/a | Involved in the study |
|---|---|
| ☒ | ☐ ChIP-seq |
| ☒ | ☐ Flow cytometry |
| ☒ | ☐ MRI-based neuroimaging |

