## [Peer Review File · Nature Ecology & Evolution]

Peer Review Information

Journal: Nature Ecology & Evolution

Manuscript Title: Mechanistic neutral models show that sampling biases drive the apparent explosion of early tetrapod diversity

Corresponding author name(s): Emma M. Dunne

Editorial Notes:

Reviewer Comments & Decisions:

Decision Letter, initial version:

21st April 2022

Dear Emma,

Your manuscript entitled "Early tetrapod diversification under neutral theory" has now been seen by three reviewers, whose comments are attached. The reviewers have raised a number of concerns which will need to be addressed before we can offer publication in Nature Ecology & Evolution. We will therefore need to see your responses to the criticisms raised and to some editorial concerns, along with a revised manuscript, before we can reach a final decision regarding publication.

The reviewers find a lot to like in the manuscript, and are concerned that some of the methodological advance is undersold. However, as you'll see reviewer 3 is concerned with advance over previous work in the field, including your 2018 paper. We will need to see a concerted response to this aspect in particular in order to consider a revision.

We therefore invite you to revise your manuscript taking into account all reviewer and editor comments. Please highlight all changes in the manuscript text file.

- * Include a "Response to reviewers" document detailing, point-by-point, how you addressed each reviewer comment. If no action was taken to address a point, you must provide a compelling argument. This response will be sent back to the reviewers along with the revised manuscript.
- * If you have not done so already please begin to revise your manuscript so that it conforms to our Article format instructions at <http://www.nature.com/natecolevol/info/final-submission>. Refer also to any guidelines provided in this letter.
- * Include a revised version of any required reporting checklist. It will be available to referees (and, potentially, statisticians) to aid in their evaluation if the manuscript goes back for peer review. A

2revised checklist is essential for re-review of the paper.

[REDACTED]

Nature Ecology & Evolution is committed to improving transparency in authorship. As part of our efforts in this direction, we are now requesting that all authors identified as 'corresponding author' on published papers create and link their Open Researcher and Contributor Identifier (ORCID) with their account on the Manuscript Tracking System (MTS), prior to acceptance. ORCID helps the scientific community achieve unambiguous attribution of all scholarly contributions. You can create and link your ORCID from the home page of the MTS by clicking on 'Modify my Springer Nature account'. For more information please visit www.springernature.com/orcid.

[REDACTED]

Reviewer expertise:

Reviewer #1: macroevolution and mass extinctions

Reviewer #2: quantitative palaeobiology

Reviewer #3: modelling ecological theory and palaeontology

Reviewers' comments:

Reviewer #1 (Remarks to the Author):

Dunne et al. present a new manuscript exploring whether raw, observed early patterns of tetrapod

2diversification are real or simply sampling artifacts, focusing on the validity increase in diversity attributed to habitat fragmentation during the Carboniferous Rainforest Collapse. In this, they fit mechanistic models based on neutral theory and different habitat loss scenarios to both the fossil record and paleographic maps.

I find this new approach to be really exciting, in that it takes into account important but often ignored fossil distribution data and provides a new way to test and tease apart the influence of biogeography and (geographic and temporal) sampling on diversification under different scenarios. It will be adapted to test a lot of different macroevolutionary hypotheses that were previously proposed on the basis of extrapolations from paleomaps, written narratives of environmental change, time series and phylogenies (like the CRC).

Of course, this paper provides a strong challenge to the major hypothesis that the CRC triggered tetrapod diversification. That's a solid proof of concept to the non-tetrapod worker such as myself but will definitely rewrite some reviews and textbooks. I recommend publication as is (and I rarely do that!)

Reviewer #2 (Remarks to the Author):

This is a cool study, there are just a few places where I think clarification would enhance your argument, increase the impact of this work, and improve understanding for the average paleo/ecology reader.

The biggest critique I have is in the way you've set up the problem and your approach in the introduction. I don't disagree with any of it, I just think it could be better explained. You're working with a lot of big ideas and pulling them together in a new way, so it's worth making sure all of the connections are explicitly stated. Essentially, there's a lot of proverbial ground to cover in not a lot of space, my main recommendation is to tighten things up so that the argument/logic flows a bit better. I have a couple of specific points in this regard, questions that I have, which I think I can answer because I work with neutral theory, but which might hinder comprehension for someone who doesn't:

- Many of the ecological terms you use throughout the manuscript (in particular: neutral theory; alpha, beta, gamma diversity) have either a few different actual meanings in use or several popular connotations. It's important to define what exactly you're taking each of these to mean so that you aren't relying on the reader's (possibly differing) interpretations.

- The manuscript starts out with a discussion of sampling biases and then quickly jumps to neutral theory with diversity partitioning mentioned as an aside, when actually it seems quite important as it forms the foundation of the figures. Comments for the introduction: First, is there any way to discuss sampling biases in fewer sentences? The way the introduction is structured, it seems like the purpose of this work is to demonstrate that neutral theory is a method for sample standardization, but your results are so much more impactful than that. Second, neutral theory is a mechanistic model to explain community assembly. I don't disagree that it can be used to address spatial gaps in sampling of the record, but that is not what it was formulated to do. What is the justification for making this connection? Why use neutral theory over any other mechanistic model? The logic behind this decision

3needs to be more explicitly justified. Finally, diversity partitioning needs to be introduced more substantially than a parenthetical about gamma diversity. Your figures are divided into alpha, beta, and gamma, which is not an unexpected thing to do with neutral theory, but you need to explain why. Also, it is the foundation for the important things you are saying about habitat fragmentation at the CRC. You mention island biogeography, which is sort of related but not quite. Explaining all of this thoroughly is important for understanding the context in which you're interpreting your results.

- How much of the discrepancy in diversity patterns at the CRC are due to purely sampling vs hierarchical spatial scaling, i.e., how do alpha, beta, and gamma relate? A full treatment of this issue is perhaps outside the scope of the manuscript, but acknowledging this caveat is relevant to your interpretation of diversity patterns. There's a ton of paleo and modern ecology literature on this. From the paleo side, start with: Sepkoski. 1988. Alpha, beta, or gamma: where does all the diversity go? And delve into papers that cite it. For ecology, Ricklefs and Rabosky have a ton of papers that are a good place to start. Also, Patzkowsky 2017 in Annual reviews is good for an overview of regional diversity and spatial patterns.

-Beta diversity is your best metric/evidence for endemism. More beta diversity = more turnover between localities, which means more endemics, i.e., more species that are unique to only 1 locality. Beta diversity actually increases for all of your amniote scenarios, but it's pretty interestingly variable for the amphibians. The interpretations as written are also unclear in regards to how you're using these metrics to test what happened at the CRC. I'm not sure if there's anything actually wrong with what's been done, rather I think this reflects just general sparseness in the discussion of alpha, beta, and gamma. A more thorough treatment of additive diversity partitioning throughout the manuscript will help clarify these interpretations and better ground them in the ecological theory that you're using.

Other more minor comments:

- Line 100: why is this expected?
- Line 102: just amniotes or amphibians also?
- Line 181: there's a typo in the punctuation of the internal citation
- Lines 210-214: does fewer coal deposits indicate less rainforest habitat? Sort of like the common cause hypothesis in marine paleobiology (see Peters and Heim 2011)
- Mention that 4 scenarios were simulated (line 354), but only 3 are discussed and presented in figure 1
- I'm not sure the upscaling total number of individuals makes sense in the context of the spatially explicit model. In neutral theory and island biogeography, habitat area should directly influence richness and number of individuals. Neutral theory in particular, has an assumption that all available space is filled. Without changing area, I'm not sure how to upscale or downscale in a way that makes sense. Or are you just changing sampling intensity? If so, I'm a bit confused about how this was determined/modelled in the unscaled results.
- Shading on the figures is not visible in pdf, but it is visualizable in word document. Not sure what to suggest here, just something to be aware of and check when submitting final versions
- The colors in figure 3 are difficult to distinguish

Reviewer #3 (Remarks to the Author):

Dunne and colleagues use neutral simulations to argue that sampling intensity alters reconstructed diversity trends (151) and use spatially explicit neutral theory to explore the mechanisms of why. The case study applies the approach to the Carboniferous Rainforest Collapse, overturning results from a previous study (Sahney et al. 2010) in an apparently qualitatively similar way to previous work by the authors (Dunne et al. 2018).

The approach is interesting and I absolutely welcome attempts to use state-of-the-art theory to bridge to palaeontological questions where sampling issues are profound. Nevertheless, as currently composed, I think the manuscript has a few substantial issues to resolve.

The novelty appears overstated - the abstract (23-25) claims that these simulations overturn previous work, but the same authors reported results that had already achieved this in 2018 (line 59-61, "the opposite of the pattern recovered by Sahney et al."). The scenarios are set up to "directly [test] the theory of Sahney et al. 2010 that endemism, driven by fragmentation, is the cause of tetrapod diversity increases post-CRC" (117-118) but "the results of Dunne et al. (2018) also indicate that fragmentation of the rainforest actually promoted the diversification" (60-61) as "the opposite of the pattern recovered by Sahney" (60). "The findings of the current paper support the previous assessment" (162-164).

The claim of a "novel direction" (239-240) is better justified in the methods sections. The background to the development of the approach is excellent (page 10), but needs to be more transparent in the main text (beyond the broader citations of similar work in the field at the bottom of page 3). Lines 367-390 does a transparent job of answering some of the quantitative questions above, particularly on the comparison between scenarios and the mean actual percentage error calculations (371-372). Although four scenarios are nominated, only three are referenced on page 4 and Figure 1 and in the Methods (358-365).

I was surprised that no uncertainty bounds were given in the Figures and model comparisons. I'm not necessarily advocating confidence intervals on simulations due to their relationship with sample size, but the literal reading of the fossil record feels like it could have some uncertainty around it. We need to know whether we have a few data points moving within a credible/confidence interval, and thus where the excursions beyond those bounds are meaningful and are left with unsatisfactory sentences like "the majority of the trend is reasonably well-captured" (104-106), what does "significant" (144), "best-fitting"(138), "aligns best" (187) mean in this context? The mean actual percentage error calculations (371-372) are hidden away rather, but would you not prefer a mean squared error so the same percentage errors in positive and negative terms do not cancel out? (e.g., the mean of +10% and -10% is 0, as is the mean of +40% and -40%, but the former is a much better fit).

The wording is too ambiguous too often - 60-62 is proposed as a key novelty of this paper, so what does indicate mean from the previous one; why "as should be expected" (101)?; the quantitative comments mentioned above (104-106); densities of what (204)).

5Elsewhere, I want to know more about how the methods yield certain results - why does sampling more individuals change the emergent patterns considerably - does this mean the initial samples are not sufficient? (172-173); if the spatial coverage falls a long way short of being truly global (180), then what are the minimal requirements for these models to have stronger predictive power? Answering that question brings me back to the lack of uncertainty bounds and so asking myself how would I know?

A revision that brought more of the detail in the methods section into the main text to expand on where the novelty of the current approach lies would allow the progress made in this manuscript be more apparent. I hope these comments help the authors refine their work.

Minor Points

Erroneous comma on line 184

Thompson et al. is 2020 in the references not 2019 as 189

*****END*****

Author Rebuttal to Initial comments

Reviewer #1

Dunne et al. present a new manuscript exploring whether raw, observed early patterns of tetrapod diversification are real or simply sampling artifacts, focusing on the validity increase in diversity attributed to habitat fragmentation during the Carboniferous Rainforest Collapse. In this, they fit mechanistic models based on neutral theory and different habitat loss scenarios to both the fossil record and paleogeographic maps. I find this new approach to be really exciting, in that it takes into account important but often ignored fossil distribution data and provides a new way to test and tease apart the influence of biogeography and (geographic and temporal) sampling on diversification under different scenarios. It will be adapted to test a lot of different macroevolutionary hypotheses that were previously proposed on the basis of extrapolations from paleomaps, written narratives of environmental change, time series and phylogenies (like the CRC). Of course, this paper provides a strong challenge to the major hypothesis that the CRC triggered tetrapod diversification. That's a solid proof of concept to the non-tetrapod worker such as myself but will definitely rewrite some reviews and textbooks. I recommend publication as is (and I rarely do that!)

We sincerely thank the reviewer for their time and positive comments.

Reviewer #2

This is a cool study, there are just a few places where I think clarification would enhance your argument, increase the impact of this work, and improve understanding for the average paleo/ecology reader.

The biggest critique I have is in the way you've set up the problem and your approach in the introduction. I don't disagree with any of it, I just think it could be better explained. You're working with a lot of big ideas and pulling them together in a new way, so it's worth making sure all of the connections are explicitly stated. Essentially, there's a lot of proverbial ground to cover in not a lot of space, my main recommendation is to tighten things up so that the argument/logic flows a bit better. I have a couple of specific points in this regard, questions that I have, which I think I can answer because I work with neutral theory, but which might hinder comprehension for someone who doesn't:

We sincerely thank the reviewer for their constructive review, which has greatly improved the manuscript. We have revised the main text, focusing on the introduction, to ensure that hypotheses, concepts and analyses are more clearly explained and that connections between them are presented explicitly. As detailed in our responses below, we have also made additions that we hope have increased clarity, enhanced our arguments, and better outlined the role of neutral theory in this work.

1. Many of the ecological terms you use throughout the manuscript (in particular: neutral theory; alpha, beta, gamma diversity) have either a few different actual meanings in use or several popular connotations. It's important to define what exactly you're taking each of these to mean so that you aren't relying on the reader's (possibly differing) interpretations.

We have added Table 1, which serves as a glossary of these terms as well as additional terms that we thought would benefit from more explicit explanations (e.g. the taxonomic groups used in the analyses)

2a. The manuscript starts out with a discussion of sampling biases and then quickly jumps to neutral theory with diversity partitioning mentioned as an aside, when actually it seems quite important as it forms the foundation of the figures. Comments for the introduction: First, is there any way to discuss sampling biases in fewer sentences? The way the introduction is structured, it seems like the purpose of this work is to demonstrate that neutral theory is a method for sample standardization, but your results are so much more impactful than that.

We have revised the introduction to improve clarity and the flow of logic, as well as to make our results and their impact much clearer from the start. In particular, we have streamlined the discussion of sampling biases (see lines 30-43) so that this does not detract from the main purpose of our methodology.

2b. Second, neutral theory is a mechanistic model to explain community assembly. I don't disagree that it can be used to address spatial gaps in sampling of the record, but that is not what it was formulated to do. What is the justification for making this connection? Why use neutral theory over any other mechanistic model? The logic behind this decision needs to be more explicitly justified.

In our revision of the introduction, we have outlined our reason for choosing to implement neutral theory more clearly (see lines 46-51). There are two main reasons why to use neutral theory. The first is to maintain simplicity so that we're left with a really minimal model that incorporates sampling and spatial structure of habitats well, but nothing else except the minimal biological functions of death, movement and reproduction. The second reason is that by assuming neutrality, efficient simulation algorithms can be used: without these it would have been practically impossible to simulate every individual tetrapod in the region.

3. Finally, diversity partitioning needs to be introduced more substantially than a parenthetical about gamma diversity. Your figures are divided into alpha, beta, and gamma, which is not an unexpected thing to do with neutral theory, but you need to explain why. Also, it is the foundation for the important things you are saying about habitat fragmentation at the CRC. You mention island biogeography, which is sort of related but not quite. Explaining all of this thoroughly is important for understanding the context in which you're interpreting your results.

We have taken several steps to improve clarity around these three measures. We now include a glossary where each of these terms is individually defined (Table 1). We have also expanded the final paragraph of the introduction to better explain our methodology and added a related explanation to the figure captions for alpha, beta and gamma diversity. Finally, we have edited the figures so that they are consistent (e.g. where species richness was used, this is now 'gamma diversity'), so that clarity is maintained throughout the manuscript.

4. How much of the discrepancy in diversity patterns at the CRC are due to purely sampling vs hierarchical spatial scaling, i.e., how do alpha, beta, and gamma relate? A full treatment of this issue is perhaps outside the scope of the manuscript, but acknowledging this caveat is relevant to your interpretation of diversity patterns. There's a ton of paleo and modern ecology literature on this. From the paleo side, start with: Sepkoski. 1988. Alpha, beta, or gamma: where does all the diversity go? And delve into papers that cite it. For ecology, Ricklefs and Rabosky have a ton of papers that are a good place to start. Also, Patzkowsky 2017 in Annual reviews is good for an overview of regional diversity and spatial patterns.

We certainly agree that a full treatment of this topic unfortunately lies outside the scope of this manuscript. Nevertheless, we have added a short discussion of this important point along with the suggested references (Sepkoski and Patzkowsky) (see lines 205-208).

5. Beta diversity is your best metric/evidence for endemism. More beta diversity = more turnover between localities, which means more endemics, i.e., more species that are unique to only 1 locality. Beta diversity actually increases for all of your amniote scenarios, but it's pretty interestingly variable for the amphibians. The interpretations as written are also unclear in regards to how you're using these metrics to test what happened at the CRC. I'm not sure if there's anything actually wrong with what's been done, rather I think this reflects just general sparseness in the discussion of alpha, beta, and gamma. A more thorough treatment of additive diversity partitioning throughout the manuscript will help clarify these interpretations and better ground them in the ecological theory that you're using.

Beta diversity is indeed the best metric to detect endemism, as well as to interpret our results in the context of the original hypothesis (i.e. that endemism boosted diversity following the Carboniferous 'rainforest collapse', CRC). In all of our figures, only Figure 6 shows simulated data with temporal sampling biases fully corrected, and so only Figure 6 shows the pattern of what, based on our models, we think was actually going on with tetrapod diversity over geological time. We have expanded our discussion around Figure 6 (see lines 179-184): here the pattern is quite consistent between amniotes and amphibians, with beta diversity (and thus endemism) increasing as hypothesized before. However, beta diversity does not increase enough to counter the decreases in alpha diversity, and so the net effect of the CRC is a decrease in gamma diversity, and thus a decrease in the global number of tetrapods.

6. Other more minor comments:

- Line 100: why is this expected?

This sentence has been edited out during the restructuring of our discussion in response to previous suggestions.

- Line 102: just amniotes or amphibians also?

Yes, this has been edited to say "amniotes and amphibians" (line 93)

- Line 181: there's a typo in the punctuation of the internal citation

Thank you, this has now been corrected.

- Lines 210-214: does fewer coal deposits indicate less rainforest habitat? Sort of like the common cause hypothesis in marine paleobiology (see Peters and Heim 2011)

Indeed. We have added a sentence noting this (see lines 224-228)

- Mention that 4 scenarios were simulated (line 354), but only 3 are discussed and presented in figure 1

Thank you, this has now been corrected.

- I'm not sure the upscaling total number of individuals makes sense in the context of the spatially explicit model. In neutral theory and island biogeography, habitat area should directly influence richness and number of individuals. Neutral theory in particular, has an assumption that all available space is filled. Without changing area, I'm not sure how to upscale or downscale in a way that makes sense. Or are you just changing sampling intensity? If so, I'm a bit confused about how this was determined/modelled in the unscaled results.

The number of simulated individuals was not changed in these scenarios. What we did is what is suggested here: varying the sampling intensity rather than the number of individuals actually present (which was not changed). We have reworded the relevant part of the methods in order to make this clearer.

- **Shading on the figures is not visible in pdf, but it is visualizable in word document. Not sure what to suggest here, just something to be aware of and check when submitting final versions**

Thank you for bringing this to our attention, we will ensure to watch for any issues in subsequent versions.

- **The colors in figure 3 are difficult to distinguish**

We have revised the colors to make them clearer. In light of this comment and the one above, we have revised the colouring and shading settings in all figures (main and supplemental) to make them easier to read.

Reviewer #3

Dunne and colleagues use neutral simulations to argue that sampling intensity alters reconstructed diversity trends (151) and use spatially explicit neutral theory to explore the mechanisms of why. The case study applies the approach to the Carboniferous Rainforest Collapse, overturning results from a previous study (Sahney et al. 2010) in an apparently qualitatively similar way to previous work by the authors (Dunne et al. 2018). The approach is interesting and I absolutely welcome attempts to use state-of-the-art theory to bridge to palaeontological questions where sampling issues are profound. Nevertheless, as currently composed, I think the manuscript has a few substantial issues to resolve.

We sincerely thank the reviewer for their constructive review, which has led to significant improvements to the manuscript.

1. The novelty appears overstated - the abstract (23-25) claims that these simulations overturn previous work, but the same authors reported results that had already achieved this in 2018 (line 59-61, “the opposite of the pattern recovered by Sahney et al.”). The scenarios are set up to “directly [test] the theory of Sahney et al. 2010 that endemism, driven by fragmentation, is the cause of tetrapod diversity increases post-CRC” (117-118) but “the results of Dunne et al. (2018) also indicate that fragmentation of the rainforest actually promoted the diversification” (60-61) as “the opposite of the pattern recovered by Sahney” (60). “The findings of the current paper support the previous assessment” (162-164).

In line with the comments from another reviewer, we have revised large portions of the main text. In doing so, we have been careful to make measured statements about the novelty of our work, including in the abstract.

2. The claim of a “novel direction” (239-240) is better justified in the methods sections.

The text has been edited so that this statement is more measured. Our claim is now encapsulated in the conclusion (see lines 225-257).

3. The background to the development of the approach is excellent (page 10), but needs to be more transparent in the main text (beyond the broader citations of similar work in the field at the bottom of page 3).

As noted above, the main text has been significantly revised to enhance clarity. We are confident that these changes address this concern.

4. Lines 367–390 does a transparent job of answering some the quantitative questions above, particularly on the comparison between scenarios and the mean actual percentage error calculations (371-372).

Thank you, we have used this positive feedback to help improve the less clear sections of the manuscript.

5. Although four scenarios are nominated, only three are referenced on page 4 and Figure 1 and in the Methods (358-365).

Thank you, this has now been corrected.

6a. I was surprised that no uncertainty bounds were given in the Figures and model comparisons. I'm not necessarily advocating confidence intervals on simulations due to their relationship with sample size, but the literal reading of the fossil record feels like it could have some uncertainty around it.

We believe this comment might stem from a PDF rendering issue as uncertainty bounds are definitely present in all of the figures. We hope that the revised figures are more clear and display the certainty bounds correctly.

6b. We need to know whether we have a few data points moving within a credible/confidence interval, and thus where the excursions beyond those bounds are meaningful and are left with unsatisfactory sentences like “the majority of the trend is reasonably well-captured” (104-106), what does “significant” (144), “best-fitting”(138), “aligns best” (187) mean in this context? The mean actual percentage error calculations (371-372) are hidden away rather, but would you not prefer a mean squared error so the same percentage errors in positive and negative terms do not cancel out? (e.g., the mean of +10% and -10% is 0, as is the mean of +40% and -40%, but the former is a much better fit).

The percentage error is calculated by taking the modulus of the difference between the simulated data and actual data as a percentage of the actual data. The modulus operator means we drop any negative sign on the result so if, say the actual value is 100 and the estimated one is 90 or 110 in either case the percentage error is 10%. This also explains how it can be reasonable to calculate the mean of such errors. During our revision of the main text, we have paid attention to our use of the phrases mentioned earlier in this comment so as not to cause confusion in readers.

7. The wording is too ambiguous too often - 60-62 is proposed as a key novelty of this paper, so what does indicate mean from the previous one; why “as should be expected” (101)?; the quantitative comments mentioned above (104-106); densities of what (204)).

Lines 60-62: The sentence containing this phrase has been reworded in our overall efforts to improve clarity throughout the manuscript, so this has been resolved. Lines 104-106: These lines have also been revised. Line 204: The neutral models explored here assumed that densities of individual organisms were consistent over time - we have clarified this in the revised text.

8a. Elsewhere, I want to know more about how the methods yield certain results - why does sampling more individuals change the emergent patterns considerably - does this mean the initial samples are not sufficient? (172-173); if the spatial coverage falls a long way short of being truly global (180), then what are the minimal requirements for these models to have stronger predictive power?

Increased sampling may uncover otherwise hidden diversity or reveal the dominance of species that turn out to have increased abundances, and not that the initial samples are insufficient. Investigating the minimal requirements for these models to have stronger predictive power would indeed be interesting, but unfortunately falls outside of the scope of the present study.

9. A revision that brought more of the detail in the methods section into the main text to expand on where the novelty of the current approach lies would allow the progress made in this manuscript be more apparent. I hope these comments help the authors refine their work.

We thank the reviewer again for their constructive review. We have carefully revised the main text to improve clarity and added details to the methods section to more clearly outline our methodology.

10. Minor Points

Erroneous comma on line 184

Thompson et al. is 2020 in the references not 2019 as 189

Both of these issues have been resolved.

Decision Letter, first revision:

29th March 2023

Dear Dr. Dunne,

Thank you for submitting your revised manuscript "Mechanistic neutral models show that sampling biases drive the apparent explosion of early tetrapod diversity" (NATECOLEVOL-220315947A). It has now been seen again by the original reviewers and their comments are below. The reviewers find that the paper has improved in revision, and therefore we'll be happy in principle to publish it in Nature Ecology & Evolution, pending minor revisions to satisfy the reviewers' final requests and to comply with our editorial and formatting guidelines.

[REDACTED]

Reviewer #2 (Remarks to the Author):

I thank the authors for their attention to my comments and those of the other reviewers. They have done an excellent job addressing them. As such, this is a much-improved manuscript. The context of their study, their results, and their interpretations are substantially clearer. I have no further changes to suggest and am excited to see this published.

I did find 2 typos:

In lines 298 and 300, the citation footnote replaces letters in the word that it's next to

In line 592-593 there is a repeat sentence

Reviewer #3 (Remarks to the Author):

I liked the first version of this manuscript; I like the revision more and thank the authors for engaging

15constructively with my comments.

I have a few minor (often very minor) outstanding issues

I think the “Investigating the minimal requirements for these models to have stronger predictive power would indeed be interesting, but unfortunately falls outside of the scope of the present study” response is not unfair; including a caveat sentence in the paragraph from lines 256-260 feels like a better response. The broader argument of this manuscript is towards more use of mechanistic models in deep time, but they will only get us so far where spatial coverage is not sufficient, which might lead us to niche differentiation and the other topics on 256-260.

79-81: yes, but assumes we are propagating errors appropriately (suggest adding the clarification as it is relevant to the result on 188-190)

107-112: feels like a contradiction here across the two sentences; clarify.

142: define “close”

151: given the run-on from the previous paragraph, is scenario C therefore the same-best-fitting scenario?

298, 300: something has gone wrong with the referencing

367: insert “eight stratigraphic” before intervals to clarify

417: presumably sampled with replacement? Insert clarification for completeness.

Our ref: NATECOLEVOL-220315947A

6th April 2023

16Dear Dr. Dunne,

Thank you for your patience as we've prepared the guidelines for final submission of your Nature Ecology & Evolution manuscript, "Mechanistic neutral models show that sampling biases drive the apparent explosion of early tetrapod diversity" (NATECOLEVOL-220315947A). Please carefully follow the step-by-step instructions provided in the attached file, and add a response in each row of the table to indicate the changes that you have made. Please also check and comment on any additional marked-up edits we have proposed within the text. Ensuring that each point is addressed will help to ensure that your revised manuscript can be swiftly handed over to our production team.

****We would like to start working on your revised paper, with all of the requested files and forms, as soon as possible (preferably within two weeks). Please get in contact with us immediately if you anticipate it taking more than two weeks to submit these revised files.****

In recognition of the time and expertise our reviewers provide to Nature Ecology & Evolution's editorial process, we would like to formally acknowledge their contribution to the external peer review of your manuscript entitled "Mechanistic neutral models show that sampling biases drive the apparent explosion of early tetrapod diversity". For those reviewers who give their assent, we will be publishing their names alongside the published article.

Nature Ecology & Evolution offers a Transparent Peer Review option for new original research manuscripts submitted after December 1st, 2019. As part of this initiative, we encourage our authors to support increased transparency into the peer review process by agreeing to have the reviewer comments, author rebuttal letters, and editorial decision letters published as a Supplementary item. When you submit your final files please clearly state in your cover letter whether or not you would like to participate in this initiative. Please note that failure to state your preference will result in delays in accepting your manuscript for publication.

Cover suggestions

As you prepare your final files we encourage you to consider whether you have any images or illustrations that may be appropriate for use on the cover of Nature Ecology & Evolution.

Covers should be both aesthetically appealing and scientifically relevant, and should be supplied at the best quality available. Due to the prominence of these images, we do not generally select images

17featuring faces, children, text, graphs, schematic drawings, or collages on our covers.

Nature Ecology & Evolution has now transitioned to a unified Rights Collection system which will allow our Author Services team to quickly and easily collect the rights and permissions required to publish your work. Approximately 10 days after your paper is formally accepted, you will receive an email in providing you with a link to complete the grant of rights. If your paper is eligible for Open Access, our Author Services team will also be in touch regarding any additional information that may be required to arrange payment for your article.

Please note that *Nature Ecology & Evolution* is a Transformative Journal (TJ). Authors may publish their research with us through the traditional subscription access route or make their paper immediately open access through payment of an article-processing charge (APC). Authors will not be required to make a final decision about access to their article until it has been accepted. [Find out more about Transformative Journals](https://www.springernature.com/gp/open-research/transformative-journals)

Authors may need to take specific actions to achieve [compliance with funder and institutional open access mandates](https://www.springernature.com/gp/open-research/funding/policy-compliance-faqs). If your research is supported by a funder that requires immediate open access (e.g. according to [Plan S principles](https://www.springernature.com/gp/open-research/plan-s-compliance)) then you should select the gold OA route, and we will direct you to the compliant route where possible. For authors selecting the subscription publication route, the journal's standard licensing terms will need to be accepted, including [self-archiving-and-license-to-publish](https://www.nature.com/nature-portfolio/editorial-policies/self-archiving-and-license-to-publish). Those licensing terms will supersede any other terms that the author or any third party may assert apply to any version of the manuscript.

[REDACTED]

[REDACTED]

Reviewer #2:

Remarks to the Author:

I thank the authors for their attention to my comments and those of the other reviewers. They have done an excellent job addressing them. As such, this is a much-improved manuscript. The context of their study, their results, and their interpretations are substantially clearer. I have no further changes to suggest and am excited to see this published.

I did find 2 typos:

In lines 298 and 300, the citation footnote replaces letters in the word that it's next to

In line 592-593 there is a repeat sentence

Reviewer #3:

Remarks to the Author:

I liked the first version of this manuscript; I like the revision more and thank the authors for engaging constructively with my comments.

I have a few minor (often very minor) outstanding issues

I think the "Investigating the minimal requirements for these models to have stronger predictive power would indeed be interesting, but unfortunately falls outside of the scope of the present study" response is not unfair; including a caveat sentence in the paragraph from lines 256-260 feels like a better response. The broader argument of this manuscript is towards more use of mechanistic models in deep time, but they will only get us so far where spatial coverage is not sufficient, which might lead us to niche differentiation and the other topics on 256-260.

79-81: yes, but assumes we are propagating errors appropriately (suggest adding the clarification as it is relevant to the result on 188-190)

107-112: feels like a contradiction here across the two sentences; clarify.

142: define "close"

151: given the run-on from the previous paragraph, is scenario C therefore the same-best-fitting scenario?

298, 300: something has gone wrong with the referencing

19367: insert "eight stratigraphic" before intervals to clarify

417: presumably sampled with replacement? Insert clarification for completeness.

Author Rebuttal, first revision:Reviewer #2:

I thank the authors for their attention to my comments and those of the other reviewers. They have done an excellent job addressing them. As such, this is a much-improved manuscript. The context of their study, their results, and their interpretations are substantially clearer. I have no further changes to suggest and am excited to see this published.

I did find 2 typos:

In lines 298 and 300, the citation footnote replaces letters in the word that it's next to
In line 592-593 there is a repeat sentence

Sincere thanks to the reviewer for their feedback and for catching these typos. They have now been corrected.

Reviewer #3

I liked the first version of this manuscript; I like the revision more and thank the authors for engaging constructively with my comments.

Sincere thanks to the reviewer for this positive feedback.

I have a few minor (often very minor) outstanding issues

I think the “Investigating the minimal requirements for these models to have stronger predictive power would indeed be interesting, but unfortunately falls outside of the scope of the present study” response is not unfair; including a caveat sentence in the paragraph from lines 256-260 feels like a better response.

This caveat has been added to the final section of the discussion, which points to avenues of future research (lines 240-241)

The broader argument of this manuscript is towards more use of mechanistic models in deep time, but they will only get us so far where spatial coverage is not sufficient, which might lead us to niche differentiation and the other topics on 256-260.

We hope that this sentiment is covered by the changes made to the lines described in the response above (i.e. lines 240-241)

79-81: yes, but assumes we are propagating errors appropriately (suggest adding the clarification as it is relevant to the result on 188-190)

To clarify this point and present a more neutral perspective on the power of our method, the sentence which was previously: "We estimate the true unbiased view of tetrapod diversity..." has been changed to "We estimate the trends in tetrapod diversity..." (lines 77-78)

107-112: feels like a contradiction here across the two sentences; clarify.

These sentences have been checked for clarity and edited to remove the contradiction (lines 103-108)

142: define "close"

This sentence (previously: "the trend...over time closely follows the changes in global diversity") has been changed to "the trend...over time *tracks* the changes in global diversity" and refers to Figure 6 (lines 144-145)

151: given the run-on from the previous paragraph, is scenario C therefore the same-best-fitting scenario?

No, the key distinction here is that the best-fitting scenario was adapted separately for amphibians and amniotes (20% random habitat loss for amniotes and a pristine landscape for amphibians; stated on lines 131-132).

298, 300: something has gone wrong with the referencing

Thank you for spotting this, these instances have been corrected (lines 278-280)

367: insert "eight stratigraphic" before intervals to clarify

Done (line 347)

417: presumably sampled with replacement? Insert clarification for completeness.

Done (line 397)

Final Decision Letter:

20th June 2023

Dear Emma,

We are pleased to inform you that your Article entitled "Mechanistic neutral models show that sampling biases drive the apparent explosion of early tetrapod diversity", has now been accepted for publication in Nature Ecology & Evolution.

Over the next few weeks, your paper will be copyedited to ensure that it conforms to Nature Ecology and Evolution style. Once your paper is typeset, you will receive an email with a link to choose the appropriate publishing options for your paper and our Author Services team will be in touch regarding any additional information that may be required

You will not receive your proofs until the publishing agreement has been received through our system

Due to the importance of these deadlines, we ask you please us know now whether you will be difficult to contact over the next month. If this is the case, we ask you provide us with the contact information (email, phone and fax) of someone who will be able to check the proofs on your behalf, and who will be available to address any last-minute problems . Once your paper has been scheduled for online publication, the Nature press office will be in touch to confirm the details.

Acceptance of your manuscript is conditional on all authors' agreement with our publication policies (see www.nature.com/authors/policies/index.html). In particular your manuscript must not be published elsewhere and there must be no announcement of the work to any media outlet until the publication date (the day on which it is uploaded onto our web site).

Please note that *Nature Ecology & Evolution* is a Transformative Journal (TJ). Authors may publish their research with us through the traditional subscription access route or make their paper immediately open access through payment of an article-processing charge (APC). Authors will not be required to make a final decision about access to their article until it has been accepted. [Find out more about Transformative Journals](https://www.springernature.com/gp/open-research/transformative-journals)

24Authors may need to take specific actions to achieve [compliance](https://www.springernature.com/gp/open-research/funding/policy-compliance-faqs) with funder and institutional open access mandates. If your research is supported by a funder that requires immediate open access (e.g. according to [Plan S principles](https://www.springernature.com/gp/open-research/plan-s-compliance)) then you should select the gold OA route, and we will direct you to the compliant route where possible. For authors selecting the subscription publication route, the journal's standard licensing terms will need to be accepted, including <https://www.nature.com/nature-portfolio/editorial-policies/self-archiving-and-license-to-publish>. Those licensing terms will supersede any other terms that the author or any third party may assert apply to any version of the manuscript.

We welcome the submission of potential cover material (including a short caption of around 40 words) related to your manuscript; suggestions should be sent to Nature Ecology & Evolution as electronic files (the image should be 300 dpi at 210 x 297 mm in either TIFF or JPEG format). Please note that such pictures should be selected more for their aesthetic appeal than for their scientific content, and that colour images work better than black and white or grayscale images. Please do not try to design a cover with the Nature Ecology & Evolution logo etc., and please do not submit composites of images related to your work. I am sure you will understand that we cannot make any promise as to whether any of your suggestions might be selected for the cover of the journal.

You can generate the link yourself when you receive your article DOI by entering it here: a

25<http://authors.springernature.com/share>.

[REDACTED]

P.S. Click on the following link if you would like to recommend Nature Ecology & Evolution to your librarian <http://www.nature.com/subscriptions/recommend.html#forms>

** Visit the Springer Nature Editorial and Publishing website at www.springernature.com/editorial-and-publishing-jobs for more information about our career opportunities. If you have any questions please click here.**